# Contextual Tokenization for Graph Inverted Indices

**Pritish Chakraborty**
IIT Bombay

**Indradyumna Roy**
IIT Bombay

**Soumen Chakrabarti**
IIT Bombay

**Abir De**
IIT Bombay

Emails: {pritish, indraroy15, soumen, abir}@cse.iitb.ac.in

## Abstract

Retrieving graphs from a large corpus, that contain a subgraph isomorphic to a given query graph, is a core operation in many real-world applications. While recent multi-vector graph representations and scores based on set alignment and containment can provide accurate subgraph isomorphism tests, their use in retrieval remains limited by their need to score corpus graphs exhaustively. We introduce CORGII (Contextual Representation of Graphs for Inverted Indexing), a graph indexing framework in which, starting with a contextual dense graph representation, a differentiable discretization module computes sparse binary codes over a learned latent vocabulary. This text document-like representation allows us to leverage classic, highly optimized inverted indices, while supporting soft (vector) set containment scores. Pushing this paradigm further, we replace the classical, fixed impact weight of a 'token' on a graph (such as TFIDF or BM25) with a data-driven, trainable impact weight. Finally, we explore token expansion to support multi-probing the index for smoother accuracy-efficiency tradeoffs. To our knowledge, CORGII is the first indexer of dense graph representations using discrete tokens mapping to efficient inverted lists. Extensive experiments show that CORGII provides better trade-offs between accuracy and efficiency, compared to several baselines. Code is in: `https://github.com/structlearning/corgii`.

## 1 Introduction

Given a query graph $G_q$, a common graph retrieval task is to find, from a large corpus of $C$ graphs, graphs $G_c$ that each contain a subgraph isomorphic to $G_q$ [1]. The ranking relaxation is to find $K$ graphs that 'best' contain $G_q$, under a suitable notion of approximate subgraph containment score. This task has several applications, *e.g.*, functional group search in molecular databases [2], control-flow pattern detection in program analysis [3], semantic search in scene graphs [4], *etc*.

Graph retrieval faces two challenges. Locally, the exact subgraph isomorphism decision problem is NP-complete [5] — but this can be circumvented via suitable score approximations, even if heuristic in nature. The more pressing global challenge is that the best approximations need early cross-interaction between $G_q$ and $G_c$, leading to an impractical $\Omega(C)$ query time. Our goal is to devise a novel indexing framework to attack the global bottleneck.

**Single vs. multi-vector graph representation tradeoffs** Approximate scores for graph containment may be computed by two families of neural networks. Early methods [6, 7, 8] use a single vector to represent a whole graph, enabling efficient relevance computation in (hashable/indexable) Euclidean space, but miss fine-grained structural details. Later methods [9, 10, 11] represent a graph as a *set* of node embedding vectors and solve a form of optimal transport [9, 10] between them to get better score approximations. This parallels the shift in dense text retrieval from late-interaction or "dual encoder" or "bi-encoder" style that pools a passage into a single vector [12, 13, 14] to multi-vector representations [15, ColBERT].

**Lessons from text retrieval** Classical text retrieval used inverted indices keyed on discrete tokens or words [16, 17, 18] mapping to posting lists of document IDs, with highly optimized implementations [19, 20, 21] that are still widely used. When word embeddings and neural text encoders became popular, first-generation dense retrieval systems used single vectors to represent queries and passage,

leading to rapid adoption of approximate nearest neighbor (ANN) indexing methods [22, 23, 24, 25, 26, 27, 28, 29, 30, 31, 32, 33, 34].

Mirroring the situation with graphs, late interaction between query and passage embeddings via cosine, L2 or dot product scores leads to efficient indices, but these scores are not as accurate as those obtained by early cross-interaction, which are not directly indexable. A notable and effective compromise was separate contextualization of query and passage words, followed by a novel *Chamfer score* [35] as an indexable surrogate of cross-interaction, as in ColBERT [15] and PLAID [36]. These methods probe the ANN index once per query word, and perform extra decompression and bookkeeping steps for scoring passages. Thus, even the partial contextualization and limited cross-interaction come at a performance premium. SPLADE [37] further improves efficiency by pre-expanding documents to include extra related words, and then use a standard inverted index on these expanded documents. In case of text, contextual similarity between words provides the signals needed for expansion.

## 1.1   Our Contributions

In proposing our system, CORGII (Contextual Representation of Graphs for Inverted Indexing), our goal is to apply the wisdom acquired from the recent history of dense text retrieval to devise a scalable index for graphs to support subgraph containment queries. In doing so, we want to use a contextual graph representation that gives accurate containment scores, but also takes advantage of decades of performance engineering invested in classic inverted indices.

**Differentiable graph tokenization**   We introduce a graph tokenizer network (GTNet) which uses a graph neural network (GNN) to map each node to a structure-aware token over a latent vocabulary. At the outset, GTNet computes binary node representations, which serve as discrete tokens, thereby forming multi-vector discrete graph representations. Our approach is significantly different from existing continuous graph embedding methods [6, 7, 9, 10, 8, 11], most of which employ Siamese networks with hinge distance to learn order embeddings. In contrast, inverted indices basically implement fast sparse dot-product computation. To reconcile this gap, we use separate tokenizer networks for query and corpus graphs, but allow a symmetric distance between the matched nodes.

Prior works [9, 10] also use injective alignment. However, we find it suboptimal, due to the subtle inconsistencies introduced by its continuous relaxation. Instead, we compute the Chamfer distance [35] over discrete node representations, which supports accurate token matching, even without injective alignment. We use these tokens to build an inverted index, where each token is mapped to the posting lists of corpus graphs containing it.

**Query-aware trainable impact score**   Text queries and documents represented as pure word sets (ignoring word counts or rarity) are not as effective for retrieval as *vector space models*, where each word has a certain *impact* on the document, based on raw word frequency, rarity of the word in the corpus (inverse document frequency or IDF), etc. Thus the query and each document are turned into sparse, non-negative, real-valued vectors.

We introduce a trainable token impact score that acts as a learned analog of classical term weights like TFIDF and BM25, but for tokenized graphs instead of text. Given a query node, this score function takes as input both its continuous and discrete representation and assigns different importance weights to the same words based on the structural information, captured through the continuous representation. This enables fine-grained, query-aware scoring while maintaining compatibility with inverted indexing.

**Recall-enhancing candidate generation prior to reranking**   Identifying the best match for a query node in a corpus graph typically requires global optimization over all nodes. In contrast, inverted indexing performs independent per-node matching, often leading to false positives. To tackle this problem, we introduce a novel co-occurrence based multiprobing strategy, where, given a token, we probe other tokens with large overlap between their posting lists. Finally, we perform a thresholded aggregation, which facilitates smooth, tunable control over the trade-off curve between accuracy and efficiency.

Experiments on several datasets show that CORGII is superior to several baselines. Moreover, the design of CORGII naturally enables a smooth trade-off between query latency and ranking quality, which is critical in many applications.

## 2 Preliminaries

**Notation.** We denote $G_q = (V_q, E_q)$ as a query graph and $G_c = (V_c, E_c)$ as a corpus graph. Let $\{G_1, G_2, ..., G_C\}$ be the set $C$ corpus graphs. Each query-corpus pair $(G_q, G_c)$ is annotated with a binary label $y_{qc} \in \{0, 1\}$, where $y_{qc} = 1$ iff $G_q \subseteq G_c$. Given corpus item indices $\mathcal{C} = [C] = \{1, .., C\}$, we define the set of relevant graphs as $\mathcal{C}_{q\oplus} = \{c : | y_{qc} = 1\}$ and the set of non-relevant graphs as $\mathcal{C}_{q\ominus} = \mathcal{C} \backslash \mathcal{C}_{q\oplus}$. Assuming $|V_q| = |V_c| = m$ obtained after suitable padding, we write $\boldsymbol{A}_q, \boldsymbol{A}_c \in \{0, 1\}^{m \times m}$ as the adjacency matrices for $G_q$ and $G_c$ respectively. $[\bullet]_+ = \max\{0, \bullet\}$ is the ReLU or hinge function. $[\![\bullet]\!]$ denotes the indicator function which is 1 if predicate $\bullet$ holds and 0 if it does not.

**Subgraph isomorphism** Given $G_q$ and $G_c$, the subgraph isomorphism problem seeks to determine iff $G_q \subseteq G_c$. This is equivalent to checking whether there exists a permutation matrix $\boldsymbol{P}$ such that $\boldsymbol{A}_q \leq \boldsymbol{P}\boldsymbol{A}_c\boldsymbol{P}^\top$, which results in a coverage loss based relevance distance, defined as: $\min_{\boldsymbol{P} \in \mathcal{P}_m} [\boldsymbol{A}_q - \boldsymbol{P}\boldsymbol{A}_c\boldsymbol{P}^\top]_+$.

Computation of $\Delta(G_q, G_c)$ defined above requires solving a quadratic assignment problem (QAP), which is NP-hard. Therefore, existing works [9, 10] propose a differentiable surrogate of this QAP using an asymmetric set distance. Given $G_q$ and $G_c$, they employ a graph neural network (GNN) to compute the node embeddings $\boldsymbol{h}_q(u), \boldsymbol{h}_c(v) \in \mathbb{R}^{\dim_h}$ for $u \in V_q$ and $v \in V_c$ respectively, then collect them in $\boldsymbol{H}_q \in \mathbb{R}^{m \times \dim_h}$ and $\boldsymbol{H}_c \in \mathbb{R}^{m \times \dim_h}$, where $m$ is the number of nodes (after padding). These embeddings are fed into a Gumbel-Sinkhorn network [38, 39], which produces a doubly stochastic matrix $\boldsymbol{P}$ — a relaxed surrogate of the binary permutation matrix — thus providing an approximate, injective node alignment map. This enables an approximation of $\min_{\boldsymbol{P} \in \mathcal{P}_N} [\boldsymbol{A}_q - \boldsymbol{P}\boldsymbol{A}_c\boldsymbol{P}^\top]_+$ as the following surrogate relevance distance:

$$\Delta(G_q, G_c) = \sum_{i \in [\dim_h], u \in [m]} [\boldsymbol{H}_q - \boldsymbol{P}\boldsymbol{H}_c]_+[u, i], \quad \boldsymbol{P} \in [0, 1]^{m \times m} \text{ is doubly stochastic.} \quad (1)$$

The underlying GNN and Gumbel-Sinkhorn network is trained under the distant supervision of binary relevance labels, without any demonstration of ground truth permutation matrix $\boldsymbol{P}$. We shall regard $\boldsymbol{H}_q, \boldsymbol{H}_c$ and Eqn. (1) as references to compute final scores of qualifying candidates that survive our index probes.

**Pretrained backbone** We use an existing subgraph matching model, namely IsoNet [9], which provides a relevance distance $\Delta(G_q, G_c)$ of the form given in Eq. (1) for any query-corpus graph pair $(G_q, G_c)$. This pre-trained backbone is employed solely at the final stage to compute $\Delta(G_q, G_c)$ for ranking the retrieved candidates. Therefore, we can access $\boldsymbol{H}_q, \boldsymbol{H}_c$ and the corresponding node alignment map $\boldsymbol{P}$.

**Inverted Index** Here, we describe inverted index based retrieval in a general information retrieval setting with query $q$ and corpus objects wih IDs $\mathcal{C} = \{1, ..., C\}$. Given a vocabulary $\omega$, we represent each instance $c$ by a set of discrete tokens $\omega(c) = \{\tau^{(1)}, \tau^{(2)}, \ldots\}$, where each $\tau^{(\bullet)} \in \omega$. For a collection of corpus items $\mathcal{C}$, the inverted index maps each token in $\omega$ to the set of corpus items containing it. Specifically, each token $\tau$ is associated with a posting list $\text{PostingList}(\tau)$, which consists of all $c \in \mathcal{C}$ containing $\tau$, $i.e.$, $\omega(c) \ni \tau$. Formally, we write $\text{PostingList}(\tau) = \text{List}(\{\tau \,|\, \tau \in \omega(c)\})$. In "impact-ordered" posting lists, corpus items are sorted in decreasing order of "impact scores" that capture the importance of a word in a document. Typically, in text retrieval, impact scores are modeled using term frequency (TF) and inverted document frequency (IDF) [17]. Given a query $x_q$, we obtain the token set $\omega(q)$ and then probe the inverted index to traverse across the posting list of each token $\tau \in \omega(q)$. Finally, we retrieve candidates from all such posting lists and return a subset from them as top-$k$ candidates.

## 3 Proposed approach

We now present CoRGII: a scalable retrieval system for graph retrieval that takes advantage of decades of optimization of inverted indices on discrete tokens, and yet supports scoring and ranking using continuous node embeddings. Starting with the hinge distance (1), we propose a series of steps that adapt GNN-based contextual node embeddings toward a discrete token space, enabling us to use inverted indices. Before describing the modules of CoRGII, we outline these adaptation steps.

**GNN-based node embeddings** As described in Section 2, a (differentiable) GNN contextualizes nodes in their graph neighborhood to output $\{\boldsymbol{x}_q(u)\}$ and $\{\boldsymbol{x}_c(v)\}$. The (transportation-inspired) hinge distance between them, found effective for ranking in earlier work, is asymmetric and based

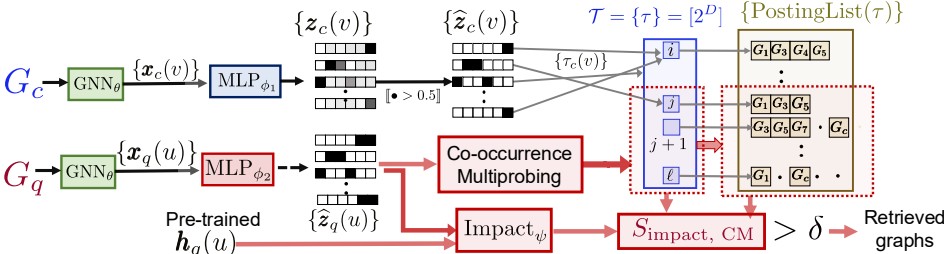

Figure 1: CORGII block diagram. Each (query, corpus) graph pair $(G_q, G_c)$ is encoded using a shared $\text{GNN}_\theta$, followed by separate MLPs ($\text{MLP}_{\phi_1}$ and $\text{MLP}_{\phi_2}$) to compute soft binary node embeddings $\boldsymbol{z}_c(v), \boldsymbol{z}_q(u) \in (0, 1)^D$. These are thresholded to obtain discrete binary codes $\widehat{\boldsymbol{z}}_c(v), \widehat{\boldsymbol{z}}_q(u) \in \{0, 1\}^D$, mapped to integer-valued latent tokens $\tau \in \mathcal{T} = [2^D]$. Corpus tokens are indexed into posting lists $\text{PostingList}(\tau)$, enabling sparse inverted indexing. During retrieval, query tokens $\tau_q(u)$ are expanded via co-occurrence–based multi-probing (CM) to select similar tokens $\mathcal{N}_b(\tau_q(u))$. Each expanded token $\tau$ contributes to the corpus score through an impact score $\text{Impact}_\psi(\tau, \boldsymbol{h}_q(u))$, producing the overall retrieval score $S_{\text{impact,CM}}(G_q, G_c)$. Graphs with score exceeding a threshold $\delta$ are shortlisted and reranked using the alignment distance $\Delta(G_q, G_c)$ (Eq. (1)).

on a (soft) permutation $\boldsymbol{P}$. These introduce two major hurdles in the way of deploying inverted indices. CORGII approximates the asymmetric, early-interaction distance (1) with an asymmetric dual encoder (late interaction) network, but based on a non-injective granular scoring function.

**Efficient differentiable (near-)tokenization** As a first step toward tokenization, we apply two (still differentiable, but distinct) networks to $\boldsymbol{x}_q(u), \boldsymbol{x}_c(v)$, ending with sigmoid activations, which take the outputs $\boldsymbol{z}_q(u), \boldsymbol{z}_c(v)$ closer to bit-vector representation of tokens. The GNN, together with these networks, are trained for *retrieval accuracy* (and not, *e.g.*, any kind of reconstruction). We also replace the permutation with a Chamfer distance [35] which brings us closer to inverted indices.

**Token discretization and indexing** Finally, we round $\boldsymbol{z}_q(u), \boldsymbol{z}_c(v)$ to 0/1 bit vectors $\widehat{\boldsymbol{z}}_q(u), \widehat{\boldsymbol{z}}_c(v)$, assigning a bit-vector token to each node. Much like text documents, a graph is now represented as a multiset of discrete tokens. With this step, we lose differentiability, but directly use an inverted index.

**Impact weights and multi-probing** All tokens should not contribute equally to match scores. Based on corpus and query workloads, we learn suitable impact weights of these (manufactured) tokens. We further optimize the performance of CORGII by designing a suitable aggregation mechanism to prune the posting lists obtained from all the query tokens. Finally, we consider one folklore and one novel means to explore the 'vicinity' of a query token, to provide a smooth trade-off between query latency and ranking accuracy.

### 3.1 Graph tokenizer network GTNet

We now proceed to describe the major components of CORGII.

The first stage of GTNet is a standard GNN similar to that described in Section 2, but here we will train it exclusively for indexing and retrieval. The GNN will share the same parameters $\theta$ across query and corpus graphs. After the GNN, we will append a multi-layer perceptron (MLP) layer with different parameters $\phi_1$ and $\phi_2$ for the query and corpus graphs.

$$\boldsymbol{z}_q(u) = \sigma(\text{MLP}_{\phi_1}(\boldsymbol{x}_q(u))) \text{ for } u \in V_q \quad \text{where, } \{\boldsymbol{x}_q(u)\} = \text{GNN}_\theta(G_q); \tag{2}$$

$$\boldsymbol{z}_c(v) = \sigma(\text{MLP}_{\phi_2}(\boldsymbol{x}_c(v))) \text{ for } v \in V_c \quad \text{where, } \{\boldsymbol{x}_c(v)\} = \text{GNN}_\theta(G_c). \tag{3}$$

**Rationale behind different MLP networks** Unlike exact graph matching, the subgraph matching task is inherently asymmetric, where $G_q \subset G_c$ does not mean $G_c \subset G_q$. To model this asymmetry, existing works [7, 9, 10] employs hinge distance $\Delta(G_q, G_c)$ (1), while sharing a a Siamese network with the same parameters for query and corpus pairs. However, such approach will preserve subgraph matching through order embeddings: $\boldsymbol{Z}_q \leq \boldsymbol{S}\boldsymbol{Z}_c$. But inverted indexing requires exact token matching, making order embeddings incompatible with token-based indexing. Therefore, we retain asymmetry through separate MLPs for queries and corpus.

**Introducing Chamfer Distance between graphs** An asymmetric Siamese network lets us replace hinge distance $[\boldsymbol{H}_q - \boldsymbol{P}\boldsymbol{H}_c]_+$ with the normed distance $\|\boldsymbol{Z}_q - \boldsymbol{P}\boldsymbol{Z}_c\|_1$, but, for the sake of indexing, we need to avoid the permutation $\boldsymbol{P}$ (whose best choice depends on both $\boldsymbol{H}_q$ and $\boldsymbol{H}_c$), so that 'document' graphs can be indexed independent of queries. Moreover, training from relevance labels

require $P$ to be modeled as a doubly stochastic soft permutation matrix (see Eq. (1)). However, its continuous nature of $P$ smears the values in $Z_q$ and $Z_c$, leading to poor discretization. Due to these reasons, we avoid the permutation, and for each query node $u$, match $z_q(u)$ to $z_c(v)$ for some corpus node $v$, *independently of other query nodes*, as opposed to the joint matching of all nodes in query corpus pairs, thus permitting non-injective mappings via the *Chamfer distance* [35]:

$$\text{Chamfer}(G_q, G_c) = \sum_{u \in V_q} \min_{v \in V_c} \|z_q(u) - z_c(v)\|_1, \tag{4}$$

Ideally, relevant graphs yield $\text{Chamfer}(G_q, G_c) = 0$, but GTNet produces approximate binary representations, making exact matches unlikely. To ensure a robust separation between the relevant and non-relevant query-corpus pairs, we seek to impose a margin of separation $m$: non-relevant graphs should differ by at least one additional bit per node compared to the relevant graphs, corresponding to a total Chamfer distance separation of $m$.

Formally, for a query graph $G_q$, and the set of relevant and irrelevant (or less relevant) corpus graphs $c_\oplus \in \mathcal{C}_{q\oplus}$ and $c_\ominus \in \mathcal{C}_{q\ominus}$, we require $\text{Chamfer}(G_q, G_{c_\oplus}) + m < \text{Chamfer}(G_q, G_{c_\ominus})$. This yields the following ranking loss optimized over parameters of GTNet, *i.e.*, $\theta, \phi_1, \phi_2$:

$$\min_{\theta, \phi_1, \phi_2} \sum_q \sum_{c_\ominus \in \mathcal{C}_{q\ominus}, c_\oplus \in \mathcal{C}_{q\oplus}} [\text{Chamfer}(G_q, G_{c_\oplus}) - \text{Chamfer}(G_q, G_{c_\ominus}) + m]_+ \tag{5}$$

Note that node embeddings $Z_q, Z_c$ still allow backprop, but are closer to "bit vectors". Moreover, the training of $\theta, \phi_1, \phi_2$ is guided not by reconstruction considerations, but purely by retrieval efficacy.

### 3.2 Discretization and inverted index

Once GTNet is trained, we compute, for each corpus graph and node therein, $z_c(v)$, and, from there, a bit vector $\widehat{z}_c(v) = [\![z_c(v) > 0.5]\!]$ as a 'hard' representation of each corpus node (and similarly from $z_q(u)$ to $\widehat{z}_q(u)$ for query nodes $u$). Given $z_c(v) \in (0, 1)^D$, this means $\widehat{z}_c(v) \in \{0, 1\}^D$, *i.e.*, each node gets associated with a $D$-bit integer. Let us call this the *token space* $\mathcal{T} = [2^D]$. Note that multiple nodes in a graph may get assigned the same token. Thus, each query graph $G_q$ and corpus graph $G_c$ are associated with *multisets* of tokens, denoted

$$\omega(G_q) = \{\!\{\widehat{z}_q(u) : u \in V_q\}\!\} \quad \text{and} \quad \omega(G_c) = \{\!\{\widehat{z}_c(v) : v \in V_c\}\!\}. \tag{6}$$

(If a graph is padded for efficient tensor operations, the tokens corresponding to padded nodes are logically excluded from the multisets. We elide this code detail for clarity.)

Conceptually, a basic inverted index is a map where the keys are tokens. Each token $\tau \in \mathcal{T}$ is mapped to the set (without multiplicity) of corpus graphs (analog of 'documents') in which it appears: $\text{PostingList}(\tau) = \{c \in \mathcal{C} : \tau \in \omega(G_c)\}$. Intuitively, the goal of minimizing the Chamfer distance (4) in the pre-discretized space corresponds, in the post-discretized space, to locating documents that have large token overlap with the query, which finally enables us to plug in an inverted index.

**Candidate generation using uniform impact** At query time, the query graph $G_q$ is processed as in (6), to obtain $\omega(G_q)$. Given the inverted index, each token $\tau \in \omega(G_q)$ is used to retrieve $\text{PostingList}(\tau)$. As a simple starting point (that we soon improve), a corpus graph can be scored as

$$S_{\text{unif}}(G_q, G_c) = \sum_{u \in V_q} [\![\widehat{z}_q(u) \in \omega(G_c)]\!]. \tag{7}$$

(If multiple nodes $u$ have the same token $\widehat{z}_q(u)$, they are counted multiple times. Belongingness in $\omega(G_c)$ is Boolean, without considering multiplicities.) These scores are used to select a subset of candidates from the whole corpus. These qualifying candidates are reranked using the (computationally more expensive) alignment-based distance $\Delta(G_q, G_c)$ (1).

### 3.3 Impact weight network

The crude unweighted score (7) has some limitations: (1) Information is lost from $H$ to $Z$ to $\hat{Z}$. Nodes with minor differences in neighborhood structure may be mapped to overlapping tokens, resulting in large candidate pools. (2) Similar to IDF, we need to discriminate against common graph motifs with poor selectivity. In our setting, the combinatorial explosion of motifs makes the estimation of motif frequencies intractable [40]. Moreover, unlike IDF in text retrieval [17], frequent structures cannot be down-weighted, as subgraph retrieval requires matching all query components, regardless of the frequency of the subgraphs.

To mitigate the above difficulties, we use a notion of token impact weight in the same spirit as in traditional text retrieval, although there are some crucial differences. We introduce an *impact weight network* $\text{Impact}_\psi : \mathcal{T} \times \mathbb{R}^{\dim_h} \to \mathbb{R}$, parameterized with $\psi$, where $\dim_h$ is the dimension of the

| (a) 2-stage training of CORGII | (b) Retrieval and reranking |
|---|---|

**(a) 2-stage training of CORGII**

1: **input:** graph corpus $\mathcal{C}$, training queries $\{G_q\}$ with relevance labels $\{y_{qc}\}$
2: ▷ *Train GTNet* ◁
3: **for** each query-corpus pair $(G_q, G_c)$ **do**
4:     Compute $\boldsymbol{Z}_q = \text{GTNet}(G_q)$ (Eq. (2))
5:     Compute $\boldsymbol{Z}_c = \text{GTNet}(G_c)$ (Eq. (3))
6:     Compute $\text{Chamfer}(G_q, G_c)$ (Eq. (4))
7: Train GTNet by minimizing margin-based ranking loss on $\text{Chamfer}(G_q, G_c)$ (Eq. (5))
8: ▷ *Train impact network* ◁
9: **for** each query-corpus pair $(G_q, G_c)$ **do**
10:     Compute $\boldsymbol{Z}_q = \text{GTNet}(G_q)$
11:     ▷ *Compute binary embeddings* ◁
12:     $\{\widehat{\boldsymbol{z}}_q(u)\} = [\![\boldsymbol{Z}_q > 0.5]\!]$
13:     ▷ *compute impact scores of all query graphs* ◁
14:     compute $S_{\text{impact}}(G_q, G_c)$ (Eq. (9))
15: Train $\text{Impact}_\psi$ network by minimizing margin-based ranking loss on $S_{\text{impact}}(G_q, G_c)$ (Eq. (10))

**(b) Retrieval and reranking**

1: **inputs:** query $G_q$, threshold $t$, pre-trained embeddings $\{\boldsymbol{h}_q(u)\}$ for $G_q$.
2: ▷ *Obtain approximate binary representation* ◁
3: Compute $\boldsymbol{Z}_q = \text{GTNet}(G_q)$ (Eq. (2))
4: ▷ *Compute binary embeddings and tokens* ◁
5: $\{\widehat{\boldsymbol{z}}_q(u)\} = [\![\boldsymbol{Z}_q > 0.5]\!]$
6: $\omega(G_q) = \text{ObtainTokenSet}(\{\widehat{\boldsymbol{z}}_q(u)\})$
7: ▷ *Compute impact weights* ◁
8: **for** each node $u \in V_q$ **do**
9:     Compute $\text{Impact}_\psi(\widehat{\boldsymbol{z}}_q(u), \boldsymbol{h}_q(u))$ (Eq. (8))
10: ▷ *Probe index using query node tokens and their impacts (with optional token expansion) and aggregate preliminary relevance scores* ◁
11: $S_{\text{impact},CM}(G_q, G_c)$ (Eq. (9))
12: ▷ *Shortlist candidates* ◁
13: $\mathcal{R}_q(\delta) = \{G_c : S_{\text{impact},CM}(G_q, G_c) \geq \delta\}$ (11)
14: rerank surviving candidates using $\Delta(G_q, G_c)$ (1)
15: **return** top-$k$ corpus graphs

Figure 2: (a) preprocessing and (b) query-time components of CORGII.

pre-trained continuous node embedding $\boldsymbol{h}_q(\bullet)$ or $\boldsymbol{h}_c(\bullet)$. We often substitute $\tau$ with $\widehat{\boldsymbol{z}}_q(u)$ in the input to $\text{Impact}_\psi$ depending on context.

**Neural architecture of** $\text{Impact}_\psi$    Network $\text{Impact}_\psi$ is implemented as a lightweight multi layer perceptron (MLP). Given input token $\tau$, presented as a binary code from $\{0, 1\}^D$, and input node embedding $\boldsymbol{h}$, we concatenate them and pass the result through a multi-layer perceptron, *i.e.*,

$$\text{Impact}_\psi(\tau, \boldsymbol{h}) = \text{MLP}_\psi(\text{concat}(\tau, \boldsymbol{h})). \tag{8}$$

Rather than count all matched tokens uniformly (7), we compute an impact-weighted aggregate:

$$S_{\text{impact}}(G_q, G_c) = \sum_{u \in V_q} \text{Impact}_\psi(\widehat{\boldsymbol{z}}_q(u), \boldsymbol{h}_q(u)) \, [\![\widehat{\boldsymbol{z}}_q(u) \in \omega(G_c)]\!] \tag{9}$$

Thus, token matches are weighted according to their learned structural importance rather than treated uniformly, enabling fine-grained, query-sensitive retrieval over the inverted index.

**Training** $\text{Impact}_\psi$    Let $\mathcal{C}_{q\oplus}$ and $\mathcal{C}_{q\ominus}$ be the relevant and non-relevant graphs for query $G_q$. Similar to Eq. (5), we encourage that $S_{\text{Impact}}(G_q, G_{c\oplus}) > S(G_q, G_{c\ominus}) + \gamma$ for $c_\oplus \in \mathcal{C}_{q\oplus}$ and $c_\ominus \in \mathcal{C}_{q\ominus}$, where $\gamma > 0$ is a margin hyperparameter. As before, this leads to a pairwise hinge loss to train $\psi$:

$$\arg\min_\psi \sum_q \sum_{c_\oplus \in \mathcal{C}_{q\oplus}} \sum_{c_\ominus \in \mathcal{C}_{q\ominus}} \left[ S_{\text{impact}}(G_q, G_{c\ominus}) - S_{\text{impact}}(G_q, G_{c\oplus}) + \gamma \right]_+. \tag{10}$$

Note that the networks described earlier, with parameters $\theta, \phi_1, \phi_2$ are frozen before training the impact parameters $\psi$. Unlike in classical inverted indices, impact weights are not associated with documents, or stored in the index. Figure 2(a) shows all the training steps of CORGII.

### 3.4 Query processing steps and multi-probing

At retrieval time, the query graph $G_q$ is first embedded using the pretrained encoder $\mathcal{E}$ to obtain node embeddings $\boldsymbol{H}_q$. The graph tokenizer GTNet then discretizes $\boldsymbol{H}_q$ into soft binary codes $\boldsymbol{Z}_q$ and later hard binary codes $\hat{\boldsymbol{Z}}$, and $\text{Impact}_\psi$ computes impact weights (if used).

Each query graph token is used to probe the inverted index. Candidate corpus graphs are retrieved by aggregating impact scores across matching tokens. Graphs with cumulative relevance scores above a tunable threshold $\delta$ form the shortlist:

$$\mathcal{R}_q(\delta) = \{c \in \mathcal{C} : S_\spadesuit(G_q, G_c) \geq \delta\}, \tag{11}$$

where $\spadesuit \in \{\text{unif}, \text{impact}\}$. Here, $\delta$ controls the trade-off between the query time and retrieval accuracy. High $\delta$ results in smaller size of $\mathcal{R}_q(\delta)$, yielding low query time, whereas a low $\delta$ gives high query time. Note that, $S_\spadesuit(G_q, G_c)$ is used only to obtain $\mathcal{R}_q$. Candidates in $\mathcal{R}_q$ are further reranked using the pretrained alignment-based 'true' distance $\Delta(G_q, G_c)$ (1). Details are in Figure 2(b).

**Limitation of single probe per query node**    We have described how candidate corpus graphs are scored using uniform and impact-weighted aggregates. In both methods, each token $\widehat{\boldsymbol{z}}_q(u)$ from the query resulted in exactly one probe into the index. Preliminary experiments suggested that a single

probe using each query token leads to lost recall, brought on partly by losing signal from continuous to bit-like node representations, and by replacing permutation-based node alignment with Chamfer score. We must discover and exploit affinities between tokens while accessing posting lists.

In the rest of this section, we explore two means to this end. The first, Hamming expansion, has already been used in the literature on locality-sensitive hashing. The second, co-occurrence expansion, is a proposal novel to CORGII.

| Term weighting | Single Probe | Hamming multiprobe (HM) | Co-occurrence multiprobe (CM) |
|---|---|---|---|
| Uniform | $S_{\text{unif}}$ | $S_{\text{unif,HM}}$ $(r = \ldots)$ | $S_{\text{unif,CM}}$ $(b = \ldots)$ |
| Impact | $S_{\text{impact}}$ | $S_{\text{impact,HM}}$ $(r = \ldots)$ | $S_{\text{impact,CM}}$ $(b = \ldots)$ (CORGII) |

Table 3: Possible combinations of term weighting and probing strategies. Default CORGII corresponds to $S_{\text{impact,CM}}$. $r$ and $b$ indicate Hamming radius for HM and number of tokens chosen for CM.

**Hamming expansion multiprobe (HM)**  While exact token matches may be adequate when query and corpus graphs are locally near-isomorphic, discretization errors and structural noise can cause relevant corpus graphs to be missed if no exact token match is found. To improve recall, we "smooth the boundaries of token bit encodings" by introducing a lightweight *token expansion* mechanism: given a query token $\tau \in \mathcal{T}$, we probe the inverted index using not only $\tau$, but also nearby tokens within a Hamming ball of radius $r$ in the binary space $\{0, 1\}^D$. Given $\widehat{z}$ and $\widehat{z}$ are the corresponding binary vectors of $\tau, \tau'$ respectively, we write $B_r(\tau) = \{\tau' : \|z - z'\|_1 \leq r\}$. $S_{\text{impact}}$ from (9) is extended, by summing over the ball, to

$$S_{\text{impact,HM}}(G_q, G_c) = \sum_{u \in V_q} \sum_{\tau \in B_r(\widehat{z}_q(u))} \text{Impact}_\psi(\tau, h_q(u)) \; [\![\tau \in \omega(G_c)]\!]. \tag{12}$$

This expansion allows retrieval of corpus graphs containing *approximate matches*, mitigating the brittleness of hard discretization without requiring dense alignment. Hamming expansion has the potential to improve recall, but there is a risk of too many false positive candidates to eliminate through expensive scoring later.

**Co-occurrence expansion multiprobe (CM)**  In classical text indexing, a token is sometimes characterized by the set of documents that mention it. Two tokens can then be compared by comparing their respective posting lists. A large overlap in these posting lists hints that the tokens have high affinity to each other. Adapting this idea to graph indexing can provide an alternative to Hamming-based affinity, which can be used either by itself, or in conjunction with Hamming-based token expansion.

For each query token $\tau \in \mathcal{T}$, we identify additional tokens $\tau'$ whose posting lists overlap significantly with the posting list of $\tau$, *i.e.*, $\text{PostingList}(\tau)$. Specifically, we define a similarity score between tokens $\tau$ and $\tau'$ as

$$\text{sim}(\tau, \tau') = \frac{|\text{PostingList}(\tau) \cap \text{PostingList}(\tau')|}{\sum_{\tau_\star \in \mathcal{T}} |\text{PostingList}(\tau) \cap \text{PostingList}(\tau_\star)|} \tag{13}$$

and expanded token set $\mathcal{N}_b(\tau) = \text{argmax}_{\tau' \in \mathcal{T}}^{(b)} \text{sim}(\tau, \tau')$, where $b$ is the number of similar tokens. Similar to $\text{Impact}_\psi$, we overload the input notation for sim where necessary. $S_{\text{impact}}$ from (9) is then updated to aggregate over this expanded neighborhood, weighted by similarity:

$$S_{\text{impact,CM}}(G_c, G_q) = \sum_{u \in V_q} \sum_{\tau \in \mathcal{N}_b(\widehat{z}_q(u))} \text{sim}(\tau, \widehat{z}_q(u)) \, \text{Impact}_\psi(\tau, h_q(u)) \; [\![\tau \in \omega(G_c)]\!]. \tag{14}$$

This way, a corpus graph $G_c$ can receive a non-zero score for a query node $u$, if any token $\tau$ in the expanded set $\mathcal{N}_b(z_q(u))$ appears in $\omega(G_c)$ — not just $z_q(u)$ itself. Table 3 lists different variants including CORGII.

## 4  Experiments

We assess the effectiveness of CORGII against several baselines on real-world graph datasets and analyze the effect of different components of CORGII. Appendix G contains additional experiments.

**Dataset**  We evaluate CORGII on four datasets from the TU benchmark suite [41]: PTC-FR, PTC-FM, COX2, and PTC-MR, which are also used existing works on graph matching [9, 10].

**Baselines**  We compare CORGII against six baselines as follows: **(1) FourierHashNet (FHN)** [42]: It is an LSH for shift-invariant asymmetric distance, computed using distance spe-

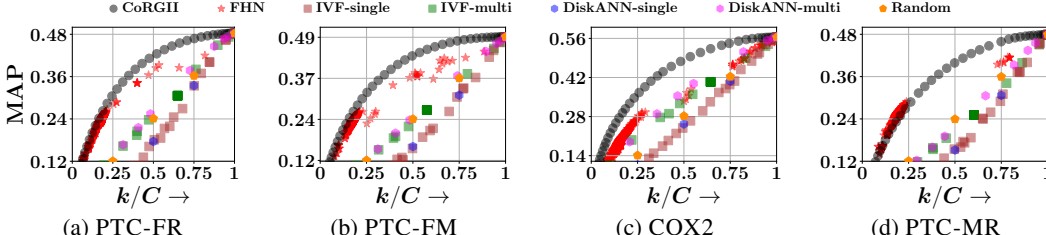

Figure 4: Tradeoff between retrieval accuracy and efficiency for CoRGII, FHN [42], IVF-single [25], IVF-multi [25], DiskANN-single [29], DiskANN [29] and Random, on 20% test queries on all four datasets. Here, retrieval accuracy is measured in terms of mean average precision (MAP) and efficiency is measured as fraction of corpus graphs retrieved ($k/C$).

cific Fourier transform. It takes single vector graph embeddings as $\boldsymbol{a}_q = \sum_{u \in V_q} \boldsymbol{h}_q(u)/|V_q|$ and $\boldsymbol{a}_c = \sum_{v \in V_c} \boldsymbol{h}_c(v)/|V_c|$ as input and builds LSH buckets for the corpus graphs $\mathcal{C}$. **(2) IVF-single**: It is a variant of IVF [25], used for single vector retrieval. Here, we build the inverted index using single vector dense corpus representations $\{\boldsymbol{a}_c\}$ and perform retrieval by probing once with $\boldsymbol{a}_q$. **(3) IVF-multi**: It is a *multi-vector* variant of IVF [25], similar to [15, ColBert]. Here, we build the index over individual node embeddings from all corpus graphs, each tagged with its parent graph ID. During retrieval, we probe the inverted index with each query node embedding and aggregating the hits by the corresponding graph IDs. **(4) DiskANN-single** [29]: It is graph-based ANN that uses HNSW index, over single vector representations. **(5) DiskANN-multi** [29]: It is a multi-vector variant of DiskANN analogous to IVF-multi. Note that IVF and DiskANN typically support $L_2$ or cosine distance. We report the results for the best performing metric. **(6) Random**: Here, we select top-$k$ items from $\mathcal{C}$ uniformly at random without replacement.

**Evaluation** Given the set of queries $\mathcal{Q}$ and the set of corpus graphs $\mathcal{C}$, we split $\mathcal{Q}$ in 60:20:20 train ($\mathcal{Q}_{\text{train}}$), dev ($\mathcal{Q}_{\text{dev}}$) and test ($\mathcal{Q}_{\text{test}}$) folds. For each query in $\mathcal{Q}_{\text{test}}$, we retrieve the corpus graphs $\mathcal{R}_q$ that are marked relevant by the corresponding model. We rerank the retrieved candidates using the pretrained ranking model (1). For each ranked list, we compute average precision (AP) and average the AP values across test queries $\mathcal{Q}_{\text{test}}$ to report on mean average precision (MAP). Given an accurate ranking model, MAP typically improves as the size of the candidate set $\mathcal{R}_q$ increases— larger sets are more likely to hit most of the relevant items, whereas smaller sets may miss many of them. To evaluate this trade-off between retrieval quality and efficiency, we measure MAP vs. the average size of the retrieved set, computed as $k = \frac{1}{|\mathcal{Q}_{\text{test}}|} \sum_{q \in \mathcal{Q}_{\text{test}}} |\mathcal{R}_q|$. To generate this trade-off curve, for FHN, we sweep over its training hyperparameters to learn multiple hashcode variants, and vary the number of hash table buckets during lookup. The implementations of IVF and DiskANN accept $k$ as input. For these baselines, we directly vary $k$ to obtain the trade-off. Appendix F report the details datasets, baselines and evaluations.

**Hyperparameters** We set $D = 10$, yielding the size of the vocabulary $|\mathcal{T}| = 2^{10}$, $b = 32$ in the expanded token set $\mathcal{N}_b(\widehat{\boldsymbol{z}}_q(u))$ used in co-occurrence based multi-probing (CM) in Eq. (14).

## 4.1 Results

**CoRGII vs. baselines** We first measure retrieval accuracy (MAP) and efficiency (inversely related to the fraction of corpus retrieved, $k/C$) across four datasets. Each curve shows how performance scales as the top-$k$ retrieved candidates vary. Figure 4 summarizes the results. We make the following observations.

**(1)** CoRGII achieves the best accuracy-efficiency trade-offs among all methods. While FHN is the strongest among the baselines, CoRGII shows particularly large gains in the high-MAP regime. For example, on the COX2 dataset, CoRGII achieves a MAP of $\sim 0.50$ at $k/C = 0.5$, whereas FHN saturates around 0.35. **(2)** Across all datasets, CoRGII achieves high MAP, very quickly at significantly lower retrieval budgets. For example, on PTC-FR, CoRGII attains a MAP of $\sim 0.36$ by retrieving less than $k = 33\%$ corpus graphs, while most baselines require more than 75% of the corpus to match that level.

**(3)** Multivector variants of IVF and DiskANN outperform their single-vector counterparts, highlighting the benefit of retaining node-level granularity. However, they still perform poorly compared to

CoRGII, largely due to their reliance on symmetric distance functions, which are unsuitable for subgraph matching — a task inherently asymmetric in nature. Single-vector variants perform the worst, sometimes even below the Random baseline, due to both the coarse nature of single-vector representations and use of symmetric distance.

**Benefits of co-occurrence based multi-probing** Here, we analyze the effect of co-occurence based multiprobing (CM) strategy (14), by comparing it with a traditional Hamming distance-based multiprobing variant (HM) (12) (Table 3 2nd vs. 3rd column, second row).

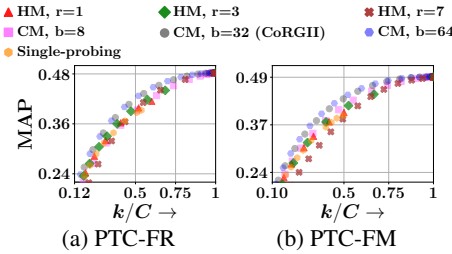

Figure 5: Multiprobing: CM vs. HM

Figure 5 shows the results for HM and CM, with different values $r$ and $b$, and single probing strategy. We make the following observations. **(1)** Single probing fails to span the full accuracy-efficiency trade-off curve across all datasets. The retrieved set is noticeably sparse at $k/C \geq 0.50$. These results highlight the necessity of multiprobing to achieve sufficient candidate coverage across varying levels of retrieval selectivity. **(2)** CM consistently achieves better trade-off than the corresponding variant of HM and single-probing strategy, while smoothly spanning the full range of retrieval selectivity. As $b$ increases, its performance improves consistently but with diminishing gains, saturating beyond $b = 32$. This indicates that a moderate number of co-occurrence-based token expansions suffices to approach near-exhaustive token expansion performance. **(3)** HM retrieves a broader range of candidates and spans the selectivity axis more effectively than the base impact score. However, as the Hamming radius $r$ increases, the expansion becomes increasingly data-agnostic, ignoring semantic alignment from $\text{Impact}_\psi$. This leads to degraded MAP at large $r$.

**Effect of impact weighting network** Next, we analyze the effect of impact weight network, by comparing with the variants of our model for both co-occurrence based multiprobing (CM, Eq. (14)) and Hamming distance based multiprobing (HM, Eq. (12)). This results in four models whose scores are $S_{\text{unif,HM}}$, $S_{\text{Impact,HM}}$, $S_{\text{unif,CM}}$ and $S_{\text{Impact,CM}}$ (CoRGII). Figure 7 summarizes the results. We observe that addition of impact weighting network improves the quality of trade-off, with significant performance gains observed for co-occurrence based multiprobing.

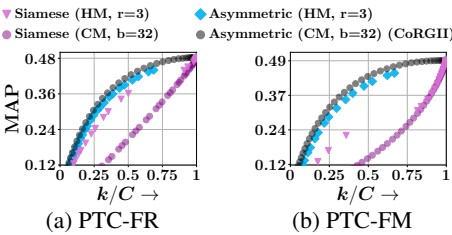

Figure 7: Ablation on impact weight network

**Siamese vs Asymmetric architecture** As discussed in Section 3.1, GTNet employs an asymmetric network architecture, which enables exact token matching while capturing the inherent asymmetry of subgraph matching. Here, we investigate its benefits by comparing against a Siamese variant of GTNet, which shares the same MLP across query and corpus graphs. Figure 9 summarizes the results for both co-occurrence based multiprobing (Eq. (14)) and Hamming based multiprobing (Eq. (12)) We highlight the following key observations: **(1)** The asymmetric variant of GTNet consistently outperforms the Siamese variant for both HM and CM. The performance boost is strikingly high for CM. **(2)** When using the asymmetric network, CM gives notable improvements over HM. However, for the Siamese variant, CM performs poorly on both PTC-FR and PTC-FM, while HM also suffers significantly on PTC-FM. This contrast highlights the importance of architectural asymmetry, especially for effective co-occurrence-based token matching.

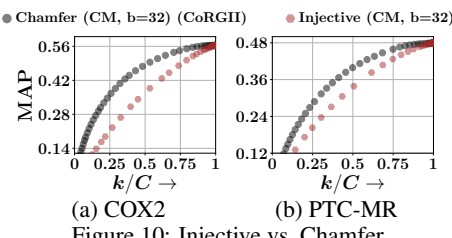

Figure 9: Siamese vs Asymmetric architecture

**Chamfer distance vs injective mapping** Chamfer distance provides a non-injective mapping. Here, we compare its performance against traditional graph matching distance with injective mapping, *i.e.*, $\|\boldsymbol{Z}_q - \boldsymbol{P}\boldsymbol{Z}_c\|_1$, where $\boldsymbol{P}$ is a soft permutation (doubly stochastic) matrix. Figure 10 shows that Chamfer distance outperforms injective alignment-based graph matching. This

Figure 10: Injective vs. Chamfer

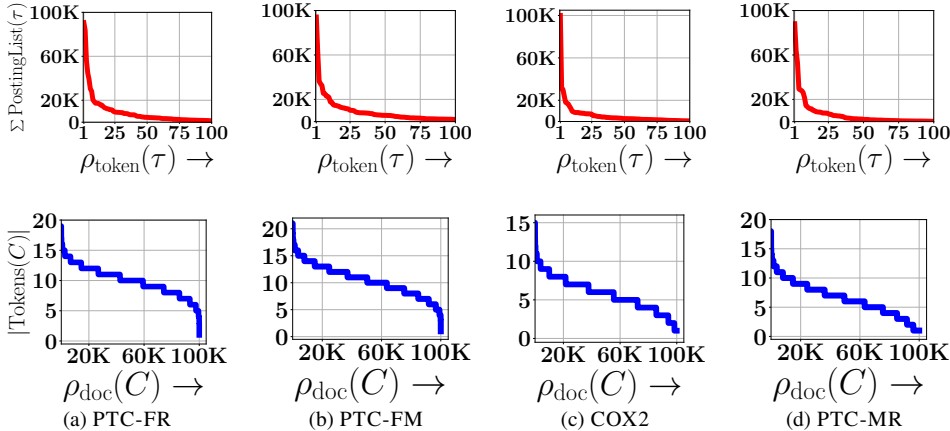

Figure 11: Top row: Posting list length vs descending token rank. Bottom row: Number of unique tokens vs descending document rank. $\rho_{\text{token}}(\tau)$ represents the rank of the token $\tau$ when sorted in descending order of posting list lengths. $\rho_{\text{doc}}(C)$ is the rank of the document $C$ when sorted in descending order of unique token count.

is because injective mappings tightly couple corpus embeddings $\boldsymbol{Z}_c$ with query embeddings $\boldsymbol{Z}_q$ preventing effective inverted indexing.

**Token rank vs Document frequency**   We rank each token in the vocabulary by the length of its posting list, $\sum \text{PostingList}(\tau)$, in descending order. Similarly, we rank each document by the number of unique tokens it contains, i.e., $|\text{Tokens}(C)|$.

In the top row of Figure 11, we plot the posting list lengths of tokens by rank. A small number of high-frequency tokens are associated with nearly all documents in the whole corpus, while the vast majority of tokens have short posting lists. The distribution exhibits a steep drop-off with rank, reminiscent of a Zipfian pattern. Inverted indexes are expected to be efficient in precisely these settings. In the second row, we plot the 'fill' of documents against documents ranked by their fills. A similar decay trend is observed, showing that most documents have a small number of tokens with non-zero impacts. Appendix G.4 contains more results.

## 5   Conclusion

We proposed CORGII, a scalable graph retrieval framework that bridges highly accurate late interaction query containment scoring with the efficiency of inverted indices. By discretizing node embeddings into structure-aware discrete tokens, and learning contextual impact scores, CORGII enables fast and accurate retrieval. Experiments show that CORGII consistently outperforms several baselines. Our work opens up several avenues of future work. It would be interesting to incorporate richly attributed graphs, capturing temporal dynamics in evolving corpora, learning adaptive token vocabularies, and exploring differentiable indexing mechanisms for end-to-end training. Another avenue is to integrate CORGII into large retrieval-augmented systems that require structured subgraph reasoning at scale.

## 6   Acknowledgements

Pritish acknowledges funding from the Qualcomm Innovation Fellowship. Indradyumna acknowledges funding from the Google PhD Fellowship and the Microsoft Research India PhD Award. Abir acknowledges funding from grants given by Amazon and Google, and the Bhide Family Chair Endowment Fund. Soumen acknowledges funding from Amazon and IBM, and the Halepete Family Chair Fund.

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

# Appendix

## A  Broader Impact

Graph retrieval is a key enabler in many real-world domains where structured relationships are central. Our work on CORGII contributes to subgraph-based retrieval, offering benefits across a range of applications:

- **Drug discovery and molecular search:** Efficient subgraph containment enables rapid screening of compounds containing functional motifs, aiding in virtual screening pipelines.
- **Program analysis and code intelligence:** Retrieval over control-flow or abstract syntax graphs can improve vulnerability detection and semantic code search.
- **Scene understanding and vision-language systems:** Graph-based representations of scenes or object relationships benefit from scalable matching of structured queries.
- **Scientific knowledge extraction:** Structured retrieval over citation or concept graphs supports discovery in large corpora of scientific knowledge.

By enabling fast and accurate retrieval of graphs under substructure containment, our work has the potential to improve the scalability and responsiveness of systems that rely on structured search over large graph collections. The proposed method, CORGII, contributes toward democratizing structure-aware search by bridging discrete indexing methods with neural representations, making such systems more accessible to low-resource settings where dense model inference may be prohibitive.

While CORGII offers efficiency and interpretability advantages, like any retrieval system, it may raise concerns when applied to sensitive graph-structured data—such as personal social networks or proprietary molecular datasets—potentially risking privacy or intellectual property leakage. Moreover, since training relies on learned embeddings, there remains a possibility of inherited biases from the underlying data. Practitioners are advised to apply appropriate safeguards, including privacy-preserving techniques and fairness auditing, when deploying the system in sensitive domains.

## B  Limitations

While CORGII demonstrates strong performance in scalable graph retrieval, several aspects offer room for further improvement. We outline them below as avenues for future exploration:

- **Static token vocabulary:** Our framework relies on a fixed latent vocabulary defined by binary token length. Learning adaptive or dynamic vocabularies could improve representation flexibility and efficiency.
- **Lack of support for attributed or heterogeneous graphs:** CORGII currently operates on purely structural information. Extending the framework to incorporate rich node/edge attributes and heterogeneous graph types is a natural next step.
- **Limited handling of evolving corpora:** The inverted index assumes a static corpus. Incorporating update-friendly indexing or continual learning mechanisms would enable deployment in dynamic settings, such as codebases or scientific repositories.

Addressing these limitations can further improve the adaptability, expressivity, and deployment readiness of CORGII in diverse graph-based retrieval settings.

## C  Clarifications

In this section, we provide few clarifications as follows.

- **Sparse vs dense representations**. In text retrieval, a "sparse" index is keyed on discrete tokens. Sparse text indexes use inverted posting lists [16, 17, 18], in which the number of document (IDs) is vastly smaller than the corpus size, hence the name "sparse". In contrast, in a "dense" text index [25, 26, 27, 28, 29, 30, 31, 32, 33, 34], a text encoder converts text into continuous vectors and then they are indexed. Such dense representations are used for IVF, LSH (e.g., FHN) or DiskANN. Dense indexes are larger and slower to navigate than inverted indexes.
- **Latent token structure and token collisions**. The latent vocabulary is structured in the form of a posting list matrix $\mathbf{PL} \in \{0,1\}^{1024 \times 100K}$. Each row of this matrix represents a token and each column a corpus item. The tokens themselves are not interpretable as they usually are in text indexing. However, the expectation from our pipeline and training scheme is that distinctive

subgraph motifs will be mapped to the same or strongly correlated groups of tokens. It is possible that different nodes from different graphs are mapped onto the same latent token, based on initial dense representation characteristics. This is further expanded upon in Appendix G.4.

# D    Related Work

## D.1    A Brief History of Text-based Retrieval Architectures

Early information retrieval (IR) systems for text, relied on sparse lexical matching using bag-of-words (BoW) representations. Documents were encoded as high-dimensional sparse vectors over a fixed vocabulary, with inverted indices mapping each term to its corresponding document sets. Statistical term-weighting schemes like TF-IDF and BM25 were used to estimate relevance, prioritizing terms that were both frequent within a document yet discriminative across the corpus. Decades of research culminated in highly optimized sparse retrieval systems, such as Lucene [19] and Elasticsearch [20], which remain industry standards for lexical search.

Despite their efficiency, lexical methods are fundamentally limited by their reliance on surface-level token matching, failing to capture deeper semantic similarity. To address this, dense neural IR models were proposed, using learned embeddings–initially static (e.g., Word2Vec [43], GloVe [44]) and later contextual (e.g., BERT [45])– to encode text into compact, low-dimensional vector spaces that support semantic retrieval. As these dense representations are incompatible with inverted indices, retrieval relies on Approximate Nearest Neighbor (ANN) search techniques such as LSH [22], HNSW [23], and IVF [46], with efficient implementations provided by libraries like FAISS [25] and ScaNN [24].

Early dense retrieval systems typically compressed entire texts into single-vector representations, but this proved suboptimal for longer inputs due to over-compression, which obscures token-level distinctions. This limitation motivated a shift toward multi-vector representations, which preserve token-level information and enable more precise semantic alignment. Architectures such as Col-BERT [15] and PLAID [36] adopt late interaction mechanisms that allow scalable token-wise retrieval. However, these methods still rely on ANN search as a subroutine, which—despite its effectiveness in dense settings—is slower than inverted indexing in practice.

To bridge the gap between dense semantic modeling and efficient retrieval, recent work has revisited sparse representations through a neural lens. Sparse neural IR models seek to combine the semantic expressiveness of dense models with the scalability and efficiency of traditional inverted indices. Approaches like SPLADE [37] learn document-specific, term-weighted sparse vectors by projecting inputs onto a high-dimensional vocabulary space. Crucially, this space is not limited to surface-level input tokens; the model can activate latent or implicitly relevant terms through learned expansions, effectively enriching the document representation beyond what is explicitly present in the text. This allows for semantically-informed exact matching within classical IR frameworks, effectively narrowing the gap between dense retrieval and sparse indexing.

## D.2    A Briefer History of Neural Graph-Containment Scoring Models

Graph containment—determining whether a query graph $G_q$ is (approximately) embedded within a corpus graph $G_c$—has long been a central problem in graph-based search. Traditional methods rely on combinatorial algorithms and subgraph isomorphism solvers, which are computationally expensive and scale poorly to large graph corpora. To address this, recent neural methods propose differentiable surrogates for containment using graph neural networks (GNNs). Raj et al. [47] provides a comprehensive analysis on different design components of neural models for subgraph isomorphism.

NeuroMatch [6] introduced a Siamese GNN with a hinge loss over aggregated node embeddings, but its global pooling loses fine structural detail. IsoNet [9] addresses this by retaining node-level embeddings $\boldsymbol{H}_q, \boldsymbol{H}_c$, computing soft alignments via a Gumbel-Sinkhorn network to produce a doubly stochastic matrix $\boldsymbol{P}$, and scoring relevance using an asymmetric hinge loss $[\boldsymbol{H}_q - \boldsymbol{P}\boldsymbol{H}_c]_+$. This better models subgraph containment and achieves improved empirical performance. IsoNet++ [10] extends this with early-interaction message passing for richer local-global representation. However, both IsoNet and IsoNet++ require dense, pairwise alignment across the corpus, limiting scalability. [48, 49, 50] use similar approaches for other type of graph similarities. Other recent works, such as [11, 7, 8], model graph similarity via node-level interactions using matching networks or soft attention. While these approaches capture structural alignment to some extent, they are tailored for general-purpose similarity tasks, making them less suited for subgraph containment and less scalable to large graph corpora.

# E  Additional details about our model and training

## E.1  Pre-trained backbone

We use Isonet [9] for final scoring mechanism. IsoNet has two components: (1) a GNN and (2) a permutation network. GNN comprises of feature initialization network $F_1$; a message computation network $F_2$ and an embedding update (or combination) network $F_3$. Specifically, for the query graph $G_q$, we execute $L$ mesage passing layers as follows:

$$\boldsymbol{h}_{q,0}(u) = F_1(\text{Feature}(u)) \quad \text{for all } u \in V_q \tag{15}$$

$$\boldsymbol{h}_{q,k+1}(u) = F_3\left(\boldsymbol{h}_{q,k}(u); \sum_{v:(u,v)\in E} F_2(\boldsymbol{h}_{q,k}(u), \boldsymbol{h}_{q,k}(v))\right), \quad \text{for all } u \in V_q, k \in \{0,..,L-1\} \tag{16}$$

We use the same procedure to compute the embeddings $\boldsymbol{h}_{c,k}$ for corpus graphs. We collect these embeddings in $\boldsymbol{H}_q, \boldsymbol{H}_c \in \mathbb{R}^{m \times \dim_h}$. These embeddings are finally used fed into multilayer perceptron, followed by dot product, to obtain an affinity matrix $\text{MLP}(\boldsymbol{H}_q)\text{MLP}(\boldsymbol{H}_c)^\top$ which is then provided as input into a node alignment network to obtain $\boldsymbol{P}$. Given a temperature hyperparameter temp, this network outputs a soft-permutation matrix using Sinkhorn iterations [38].

$$\boldsymbol{P} = \text{Sinkhorn}(\text{MLP}(\boldsymbol{H}_q)\text{MLP}(\boldsymbol{H}_c)^\top/\text{temp}) \tag{17}$$

Gumbel-Sinkhorn network consists of iterative row-column normalization as follows:

$$\boldsymbol{P}_0 = \exp(\text{MLP}(\boldsymbol{H}_q)\text{MLP}(\boldsymbol{H}_c)^\top/\text{temp}) \tag{18}$$

$$\boldsymbol{P}_{t+1} = \text{RowNormalize}(\text{ColumnNormalize}(\boldsymbol{P}_t)) \quad 0 \le t \le T-1. \tag{19}$$

As $T \to \infty$ $\boldsymbol{P}_T$ approaches as doubly stochastic matrix and as temp $\to 0, T \to \infty$, the matrix $\boldsymbol{P}_T$ approaches a permutation matrix.

In our work, we set $\dim_h = 10$. Here, $F_1$ is 10-dimensional encoder; $F_2$ consists of a combination of a propagator layer with a hidden dimension of 20 and a GRU layer at the output, with final dimension 10; and $F_3$ consists of an aggregator layer with hidden dimension 20 and output dimension 10. The MLPs used in Sinkhorn network, are linear-ReLU-linear networks. Each MLP has a hidden layer of 25 dimensions, and the output is of 25 dimensions. Finally, we minimize the ranking loss to obtain the parameters of $F_1, F_2$ and $F_3$ (Eq. (15)– (16)); and MLP used in Eq. (19)

$$\sum_q \sum_{c_\oplus \in \mathcal{C}_{q\oplus}} \sum_{c_\ominus \in \mathcal{C}_{q\ominus}} \left[\Delta(G_q, G_{c_\oplus}) - \Delta(G_q, G_{c_\ominus}) + \text{Margin}\right]_+. \tag{20}$$

We used a margin of 0.5. Note that $\Delta$ is *only* used in the final stage of ranking. In Sinkhorn network, we set the number of iterations $T = 10$ and temparature 0.1.

## E.2  Details about CoRGII

**Architecture of** GTNet **and** $\text{Impact}_\psi$   The GNN in GTNet consists of same architecture as in Eqs. (15)– (16), with the same number of layers and hidden dimensions. Here, we set $\dim(\boldsymbol{x}_\bullet) = 10$. Each of the MLPs in GTNet, *i.e.*, $\text{MLP}_{\phi_1}, \text{MLP}_{\phi_2}$ in Eqs. (2) and (3) consist of a linear-ReLU-linear network with input dimension 10, hidden layer of size 64 and output dimension 10. Note that GTNet does not share any components with the pre-trained backbone.

$\text{MLP}_\psi$ used in Eq. (8) to model the impact scorer admits a similar architecture as $\text{MLP}_{\phi_1}, \text{MLP}_{\phi_2}$. It consists of a linear-ReLU-linear network with input dimension 10, hidden layer of size 64 and output dimension 10.

**Optimization and Early Stopping.**   We train both models using the Adam optimizer with a learning rate of $1 \times 10^{-3}$ and a batch size of 3000. During GTNet training, early stopping is performed at the sub-epoch level (i.e., across batches) with a patience of 30 steps and validation every 30 steps. For $\text{Impact}_\psi$, early stopping is applied at the epoch level with a maximum of 20,000 epochs and patience set to 50. Validation is conducted every epoch, with a default tolerance threshold of $5 \times 10^{-3}$. In both cases, the model is evaluated using the score function aligned with its training objective.

**Margin Hyperparameter Tuning.**   For the Chamfer-based ranking loss in Eq. (5), we experiment with margin values of $\{0.01, 0.1, 1.0, 10, 30\}$. The best-performing margins are 10 for PTC-FR and PTC-FM, and 30 for COX2 and PTC-MR. For the impact network loss in Eq. (10), tested margins

include $\{0.01, 0.1, 1.0\}$. Margins of 0.01, 0.01, 1.0, and 0.1 work best for PTC-FR, PTC-FM, COX2, and PTC-MR, respectively.

**Training Under Co-Occurrence Expansion.** During training with co-occurrence multiprobing, the token neighborhood $\mathcal{N}_b(\hat{z}_q(u))$ in Eq. (14) is replaced with the full vocabulary $\mathcal{T}$. This allows $\mathrm{Impact}_\psi$ to learn a relevance-aware importance score for every token. At retrieval time, top-$b$ tokens are selected based on $\mathrm{sim}$, using the learned impact scores.

**Reproducibility.** All experiments are run with a fixed random seed of 42 across libraries and frameworks. We leverage PyTorch's deterministic execution setting and CuBLAS workspace configuration to ensure reproducible execution.

# F Additional details about experiments

| Dataset | $\min(|V_C|)$ | $\max(|V_C|)$ | $\mathbb{E}[|V_C|]$ | $\min(|E_C|)$ | $\max(|E_C|)$ | $\mathbb{E}[|E_C|]$ |
|---|---|---|---|---|---|---|
| **PTC-FR** | 16 | 25 | 18.68 | 15 | 28 | 20.16 |
| **PTC-FM** | 16 | 25 | 18.70 | 15 | 28 | 20.13 |
| **COX2** | 16 | 25 | 19.65 | 15 | 26 | 20.23 |
| **PTC-MR** | 16 | 25 | 18.71 | 15 | 28 | 20.17 |

(a) Corpus graph statistics.

| Dataset | $\min(|V_Q|)$ | $\max(|V_Q|)$ | $\mathbb{E}[|V_Q|]$ | $\min(|E_Q|)$ | $\max(|E_Q|)$ | $\mathbb{E}[|E_Q|]$ | $\mathbb{E}[\frac{|y=1|}{|y=0|}]$ |
|---|---|---|---|---|---|---|---|
| **PTC-FR** | 6 | 15 | 12.64 | 6 | 15 | 12.41 | 0.11 |
| **PTC-FM** | 7 | 15 | 12.58 | 7 | 15 | 12.34 | 0.12 |
| **COX2** | 6 | 15 | 13.21 | 6 | 16 | 12.81 | 0.11 |
| **PTC-MR** | 6 | 15 | 12.65 | 7 | 15 | 12.41 | 0.12 |

(b) Query graph statistics and average positive-to-negative label ratio ($\mathbb{E}[\frac{|y=1|}{|y=0|}]$).

Table 12: Statistics of sampled subgraph datasets used in our experiments. Each dataset consists of 500 query graphs and 100,000 corpus graphs.

## F.1 Datasets

All experiments are performed on the following datasets: PTC-FR, PTC-FM, COX2 and PTC-MR [9, 51]. From each dataset, we extract corpus and query graphs using the sampling procedure outlined in [9], such that $|\mathcal{C}| = 100000$ and $|\mathcal{Q}| = 500$. The queryset is split such that $|\mathcal{Q}_{\mathrm{train}}| = 300$, $|\mathcal{Q}_{\mathrm{dev}}| = 100$ and $|\mathcal{Q}_{\mathrm{test}}| = 100$. Each dataset has its relevant statistics outlined in Table 12, including the minimum, maximum and average number of nodes and edges in both the corpus set and the queryset. Additionally, the table also lists the per-query average ratio of positive ground truth relationships to negative ground truth relationships.

## F.2 Baselines

We provide a detailed description of each of the baselines used in our experiments.

**FourierHashNet** It is a Locality-sensitive Hashing (LSH) mechanism designed specifically for the set containment problem [42], applied to subgraph matching. In particular, it overcomes the weaknesses of symmetric relevance measures. Earlier work employed measures such as Jaccard similarity, cosine similarity and the dot product to compute similarity between a pair of items, which do not reflect the asymmetric nature of the problem. FHN, on the other hand, employs a hinge-distance guided dominance similarity measure, which is further processed using a Fourier transform into the frequency domain. The idea is to enable compatibility with existing fast LSH techniques by leveraging inner products in the frequency domain, while retaining asymmetric notion of relevance.

We adopt the original architecture and training settings of FourierHashNet [42] without modification. The model employs 10 sampled Fourier frequencies to compute learned asymmetric embeddings, which are then optimized using a binary cross-entropy loss over embedding vectors of dimension 10. The final hash representation consists of 64-bit codes. For training, we perform a full grid sweep over the hyperparameter configurations proposed in the original work, including all specified loss weights.

To study the trade-off between retrieval accuracy and index efficiency, we vary the number of hash table buckets at query time, ranging from $2^1$ to $2^{60}$.

**IVF**   The FAISS library provides facilities for inverted file indexing (IVF) [25]. IVF clusters the corpus of vectors using a suitable quantization method. The quantization method produces centroid vectors for each corpus vector, and each centroid represents a cluster. Internally, the library stores the vectors assigned to each cluster in the form of (possibly compressed) contiguous postings lists. To search this construction, a query vector is transformed into its corresponding centroid to match with the given cluster. Depending on the number of probes argument given during search time, one may expand their search into multiple neighboring clusters. We implement single-vector and multi-vector variants of IVF.

Note that we use the `faiss.IndexFlatIP` as the quantizer and `faiss.IndexIVFFlat` as the indexer.

**DiskANN**   To tackle the challenge of having to store search indices in memory for strong recall, DiskANN introduces efficient SSD-resident indices for billion-scale datasets [29]. To this end, the authors develop a graph construction algorithm inspired by methods such as HNSW [52], but which produce more compact graphs (smaller diameter). They construct smaller individual indices using this algorithm on overlapping portions of the dataset, and then merge them into a single all-encompassing index. These disk-resident indices can then be searched using standard techniques. One of the key benefits of DiskANN is that it requires modest hardware for the construction and probing of their disk-resident indices. We implement single-vector and multi-vector variants. Note that we test against the memory-resident version of DiskANN.

We employ a graph degree of 16, complexity level of 32, alpha parameter of 1.2 during indexing. During search, we use an initial search complexity of $2^{21}$.

### F.3   Evaluation metric

We report Mean Average Precision (MAP) and the average number of retrieved candidates to characterize the efficiency–accuracy tradeoff. Retrieved candidates are reranked using the pretrained alignment model for consistent evaluation. For a query $q$ with relevant corpus set $\mathcal{C}_{q\oplus}$ and a retrieved ranking $\pi_q$, we define the average precision (AP) as

$$\frac{1}{|\mathcal{C}_{q\oplus}|} \sum_{r=1}^{|\pi_q|} \text{Prec}@r \cdot \text{rel}_q(r) \tag{21}$$

where $\text{rel}_q(r) \in \{0, 1\}$ indicates whether the $r$-th ranked item is relevant to $q$ and $\text{Prec}@r$ is the precision at rank $r$. MAP is the mean of AP over all queries. This formulation penalizes high precision with low recall, ensuring models are rewarded only when most number of relevant items are retrieved with high retrieval accuracy.

### F.4   System configuration

All experiments were conducted on an in-house NAS server equipped with seven 48GB RTX A6000 GPUs respectively. All model training is done on GPU memory. Further, the server is equipped with 96-core CPU and a maximum storage of 20TB, and runs Debian v6.1. We found that this hardware was sufficient to train CORGII.

### F.5   Licenses

Our code will be released under the MIT license. DiskANN, FAISS and FourierHashNet are all released under the MIT license.

# G Additional experiments

We present additional experimental results covering the comparison between co-occurrence-based multiprobing (CM) and Hamming multiprobing (HM), the ablation study on the impact scorer $\text{Impact}_\psi$, and the effect of using a Siamese versus asymmetric architecture in GTNet. We also include supporting analyses on posting list statistics—such as token frequency distributions and posting list co-occurrence patterns—as well as extended results for various CORGII variants.

## G.1 CM vs HM

Figure 5 in the main paper compares co-occurrence multiprobing (CM) with Hamming multiprobing (HM) on the PTC-FR and PTC-FM datasets. Here, we present additional results on the COX2 and PTC-MR datasets.

Figure 13 confirms that CORGII remains the best-performing method overall, though the gap between CM and HM narrows on PTC-MR. **(1)** On COX2, HM fails to sweep the entire selectivity axis; even with $r = 7$, it only reaches up to $k/C = 0.6$. **(2)** CM steadily improves with increasing $b$, approaching exhaustive coverage, and saturates beyond $b = 32$. **(3)** On PTC-MR, while the difference between HM ($r = 7$) and CM ($b = 32$) is less pronounced, HM still does not cover the full selectivity range. **(4)** The trends and conclusions drawn in Section 4.1 remain consistent across these datasets.

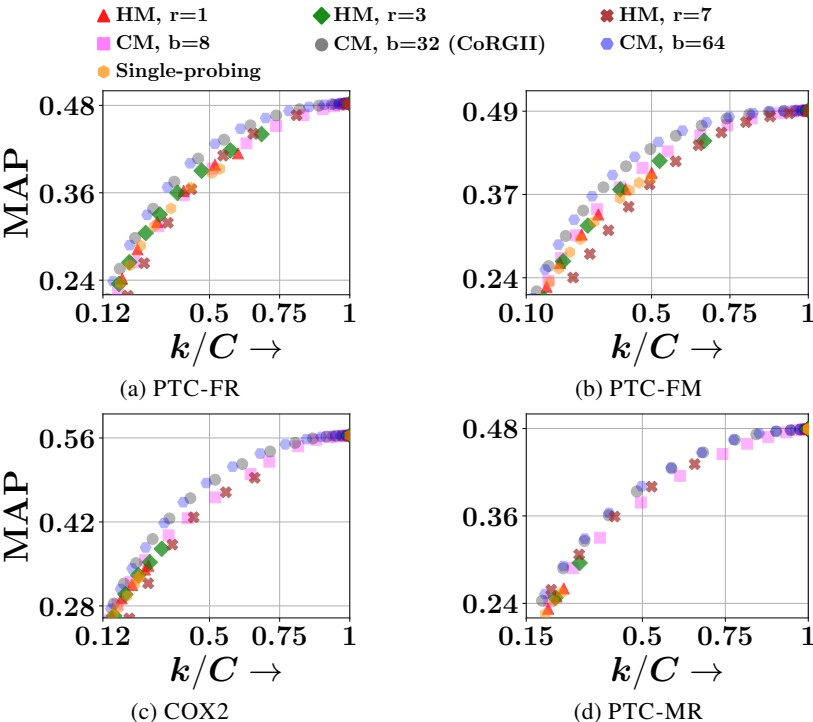

Figure 13: Comparison of Hamming expansion multiprobing (HM) against co-occurrence expansion multiprobing (CM) across four real-world datasets and across several values of $r$ (Hamming ball radius) and $b$ (number of topmost co-occurring tokens). Each plot consists of tradeoffs between selectivity ($k/C$) and MAP, for different values of $r$ and $b$. $b = 32$ is sufficient for CM to outperform HM variants. To deal with the crowding and overlapping problem in the plots, we have applied point sub-sampling on CORGII.

## G.2 Impact weight network ablation

Figure 7 in the main paper examines the impact of the weighting network on the PTC-FR and PTC-FM datasets. We now extend this analysis to COX2 and PTC-MR in Figure 14, with the following observations: **(1)** CORGII continues to outperform all other variants across the full retrieval budget spectrum on both datasets. **(2)** On COX2, uniform aggregation with Hamming multiprobing (HM) briefly approaches CORGII at low $k/C$ values, but quickly falls behind as selectivity increases. **(3)** Removing impact weights causes a significant drop in CM performance across both datasets, underscoring the value of learned token-level importance. **(4)** Uniform aggregation under CM fails to deliver competitive trade-offs, confirming that context-aware impact scoring is essential for effective retrieval.

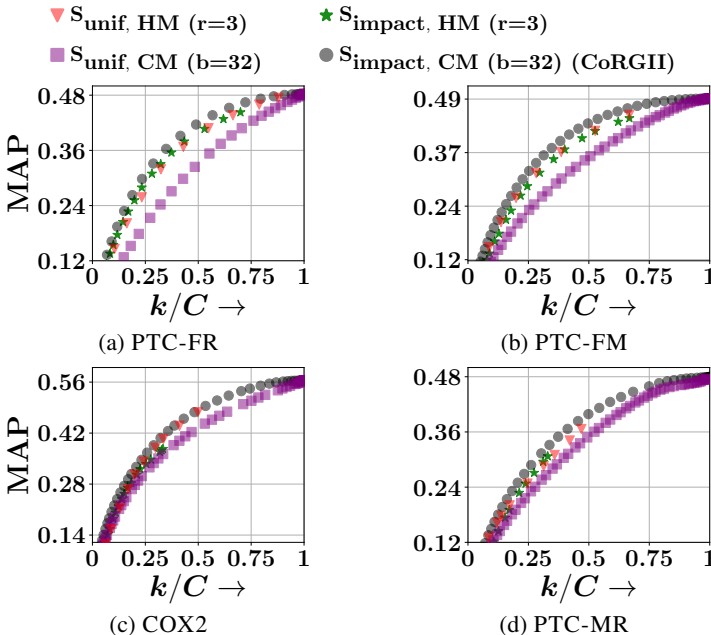

Figure 14: Effect of ablation on the impact weight network across four datasets. Each subplot compares four retrieval variants: uniform aggregation with Hamming multiprobing ($S_{\text{unif, HM}}$), uniform aggregation with co-occurrence multiprobing ($S_{\text{unif, CM}}$), impact-weighted Hamming multiprobing ($S_{\text{impact, HM}}$), and impact-weighted co-occurrence multiprobing ($S_{\text{impact, CM}}$, the default CORGII).

### G.3 Siamese vs Asymmetric networks

Figure 9 in the main paper analyzes the contribution of CORGII's asymmetric architecture on the PTC-FR and PTC-FM datasets. Figure 15 complements this analysis with results on COX2 and PTC-MR.

We observe: **(1)** CORGII consistently outperforms both Siamese variants (with CM and HM probing), reaffirming the importance of architectural asymmetry for subgraph containment. **(2)** Among the HM variants, the asymmetric network achieves a better tradeoff curve compared to the Siamese counterpart, particularly evident in the mid-selectivity range. **(3)** Despite this, HM-based variants—both asymmetric and Siamese—fail to span the full selectivity axis, highlighting the limitations of Hamming multiprobing for recall. **(4)** These results further validate the need for asymmetry in the encoder architecture to accurately reflect containment semantics under both probing schemes.

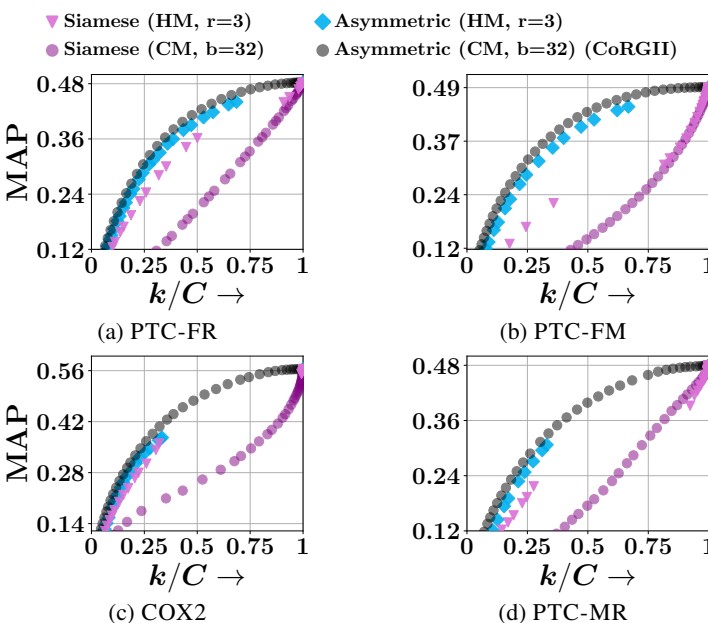

Figure 15: Ablation comparing Siamese and asymmetric network architectures under different probing strategies. Each variant combines one of two network architectures—**Siamese** (shared MLP for query and corpus) and **Asymmetric** (separate MLPs)—with one of two probing strategies: **HM** (Hamming multiprobing with radius $r = 3$) or **CM** (Co-occurrence multiprobing with $b = 32$). CORGII corresponds to the **Asymmetric + CM** configuration, shown as black circles.

## G.4 Insights into token co-occurrence

Drawing parallels from natural language and retrieval systems, the structure of posting lists and corresponding token co-occurrence statistics are of key interest. In this section, we examine how these properties vary across datasets.

**Co-occurrence statistics** Table 16 reports structural statistics of the posting list matrix across all datasets. Let the posting list matrix be $\mathbf{PL} \in \{0,1\}^{1024 \times 100K}$, where each row represents a token and each column a document. The corresponding co-occurrence matrix is defined as $\mathbf{C} = \mathbf{PL} \cdot \mathbf{PL}^{\mathsf{T}} \in \mathbb{Z}_+^{1024 \times 1024}$.

We list both the *actual rank* and the *effective rank* of $\mathbf{PL}$, the latter computed using the energy-preserving criterion from truncated singular value decomposition (SVD) [53, 54]. Let $\sigma_1, \ldots, \sigma_n$ denote the singular values of $\mathbf{PL}$. The effective rank is the smallest $K$ such that $\frac{\sum_{i=1}^{K} \sigma_i^2}{\sum_{i=1}^{n} \sigma_i^2} > \gamma$, with $\gamma = 0.95$. Since $\text{rank}(\mathbf{PL}) = \text{rank}(\mathbf{C})$ but $\text{rank}^{\text{eff}}(\mathbf{PL}) \geq \text{rank}^{\text{eff}}(\mathbf{C})$, the effective rank of $\mathbf{PL}$ serves as an informative upper bound for that of $\mathbf{C}$.

The large gap between the actual and effective rank across datasets—particularly the low effective rank—indicates that token co-occurrences lie on a low-dimensional manifold. This suggests that both $\mathbf{PL}$ and $\mathbf{C}$ are highly compressible, enabling projection onto a lower-dimensional subspace without significant loss of information. This again resembles the behavior of text corpora, where the discrete word space may be in the tens or hundreds of thousands, but a few hundred dense dimensions suffice to encode words and documents [27].

| Dataset | Actual Rank | Effective Rank |
|---------|-------------|----------------|
| PTC-FR  | 393         | 4              |
| PTC-FM  | 498         | 9              |
| COX2    | 250         | 7              |
| PTC-MR  | 232         | 3              |

Table 16: Actual and effective rank (95% SVD energy threshold) of the posting list matrix $\mathbf{PL}$ for each dataset.

## G.5 End-to-End Training vs Frozen Backbone for $S_{unif, CM}$ and $S_{unif, HM}$

A key design consideration is whether GTNet benefits from end-to-end training using its own GNN encoder, or whether comparable results can be obtained using frozen embeddings from a pretrained backbone. Figure 17 compares the performance of $S_{unif, CM}$ and $S_{unif, HM}$ under both configurations.

We observe that: **(1)** End-to-end training consistently yields better MAP–selectivity tradeoffs for both CM and HM variants, indicating that learning task-specific embeddings improves token discriminability. **(2)** The frozen backbone variant spans a wider range of selectivity values ($k/C$), suggesting looser token matching and higher recall, but at the cost of reduced precision. **(3)** End-to-end models tend to retrieve fewer candidates for the same threshold, reflecting tighter, more precise tokenization.

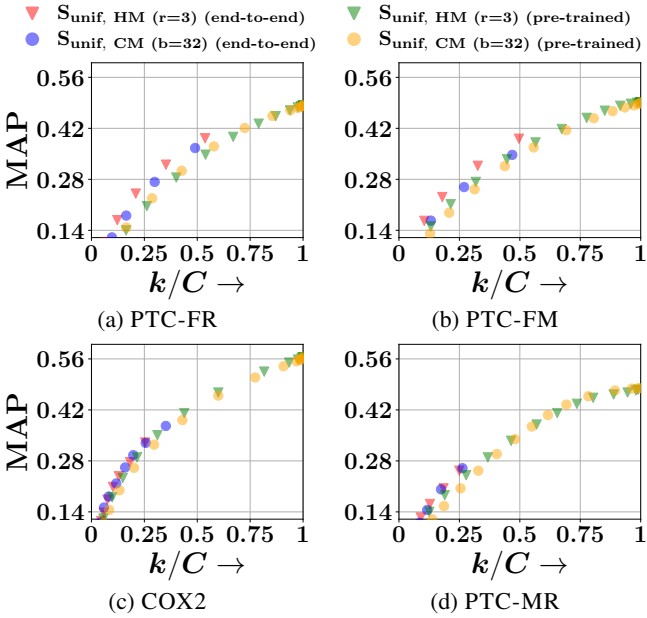

Figure 17: Comparison of end-to-end training versus using a frozen backbone for $S_{unif, CM}$ and $S_{unif, HM}$, with $r = 3$ and $b = 32$. End-to-end training refers to learning the GTNet encoder jointly, while the frozen variant reuses pretrained embeddings.

In the next set of ablations, we show that the impact network does not need pre-trained IsoNet embeddings of the given dataset to maintain superior tradeoffs.

## G.6 Using pre-trained IsoNet embeddings of different dataset as input to Impact$_\psi$

Instead of using pre-trained IsoNet embeddings $h$ on the same dataset, we use $h$ trained on the PTC-FM dataset as input to the impact network for PTC-FR, and likewise use $h$ trained on PTC-MR and apply to the impact network for COX2. Table 18 shows the efficiency in terms of fraction of graphs $k/C$ retrieved to achieve atleast a certain MAP=$m^*$. Lower $k/C$ is better and indicates higher efficiency. _ denotes places where the method is unable to achieve the target MAP.

## G.7 Using intermediate embeddings $x$ of GTNet as input to Impact$_\psi$

Next, instead of pre-trained embeddings $h$, we use $x$ as input to the impact scoring model (Impact$_\psi(\tau, x)$), (8)), where $x$ is the embedding from the GTNet GNN, (which is not pretrained and trained as a part of the CORGII scheme).

The following Table 19 shows the efficiency in terms of fraction of graphs $k/C$ retrieved to achieve atleast a certain MAP=$m^*$ for PTC-FR (first subtable) and COX2 (second subtable) datasets. Lower $k/C$ is better and indicates higher efficiency.

| $m^*$ | CoRGII | | FHN | | IVF | |
|---|---|---|---|---|---|---|
| | **current** | **($h$ trained on PTC-FM)** | **default** | **($h$ trained on PTC-FM)** | **default** | **($h$ trained on PTC-FM)** |
| 0.38 | 0.38 | 0.42 | 0.51 | 0.97 | 0.76 | 0.81 |
| 0.40 | 0.44 | 0.47 | 0.81 | 1.0 | 0.89 | 0.81 |
| 0.45 | 0.645 | 0.69 | 0.99 | 1.0 | 0.96 | 0.91 |

(a) $k/C$ values for **PTC-FR**.

| $m^*$ | CoRGII | | FHN | | IVF | |
|---|---|---|---|---|---|---|
| | **current** | **($h$ trained on PTC-MR)** | **default** | **($h$ trained on PTC-MR)** | **default** | **($h$ trained on PTC-MR)** |
| 0.36 | 0.31 | 0.27 | 0.61 | 1.0 | 0.64 | 0.57 |
| 0.38 | 0.35 | 0.29 | 0.71 | – | 0.64 | 0.81 |
| 0.40 | 0.45 | 0.37 | 0.99 | – | 0.83 | 0.81 |

(b) $k/C$ values for **COX2**.

Table 18: $k/C$ values comparing CoRGII, FHN, and IVF with and without transferring $h$ across datasets. CoRGII outperforms the baselines even when the pre-trained $h$ of a different dataset is used.

| $m^*$ | CoRGII | | FHN | DiskANN-multi | IVF-multi |
|---|---|---|---|---|---|
| | **(current)** | **(use $x$)** | | | |
| 0.38 | 0.38 | 0.37 | 0.51 | 0.87 | 0.76 |
| 0.4 | 0.44 | 0.43 | 0.81 | 0.87 | 0.89 |
| 0.45 | 0.64 | 0.62 | 0.99 | 0.93 | 0.96 |

(a) $k/C$ for **PTC-FR**.

| $m^*$ | CoRGII | | FHN | DiskANN-multi | IVF-multi |
|---|---|---|---|---|---|
| | **(current)** | **(use $x$)** | | | |
| 0.36 | 0.31 | 0.42 | 0.61 | 0.55 | 0.64 |
| 0.38 | 0.35 | 0.48 | 0.71 | 0.82 | 0.64 |
| 0.42 | 0.45 | 0.58 | 0.99 | 0.82 | 0.83 |

(b) $k/C$ for **COX2**.

Table 19: $k/C$ values comparing CoRGII (current $h$ vs. using $x$), FHN, DiskANN-multi, and IVF-multi. CoRGII outperforms each of the baselines even when not using $h$ as impact network input.

## G.8 Using pre-trained Graph Embedding Network embeddings as input to Impact$_\psi$

We pre-train a different model called Graph Embedding Network (GEN) [8], which is different from the final reranking model. Table 20 contains the results, with _ denoting places where the method is unable to achieve target MAP:

| $m^*$ | Ours | | FHN | | DiskANN-multi | | IVF-multi | |
|---|---|---|---|---|---|---|---|---|
| | **(current)** | **(pretrained GEN-embed.)** | **(current)** | **(pretrained GEN-embed.)** | **(current)** | **(pretrained GEN-embed.)** | **(current)** | **(pretrained GEN-embed.)** |
| 0.38 | 0.38 | 0.38 | 0.51 | 1.0 | 0.87 | 0.82 | 0.76 | 0.82 |
| 0.40 | 0.44 | 0.44 | 0.81 | – | 0.87 | 0.88 | 0.89 | 0.94 |
| 0.45 | 0.64 | 0.65 | 0.99 | – | 0.93 | 0.98 | 0.96 | 0.94 |

(a) $k/C$ for **PTC-FR**.

| $m^*$ | Ours | | FHN | | DiskANN-multi | | IVF-multi | |
|---|---|---|---|---|---|---|---|---|
| | **(current)** | **(pretrained GEN-embed.)** | **(current)** | **(pretrained GEN-embed.)** | **(current)** | **(pretrained GEN-embed.)** | **(current)** | **(pretrained GEN-embed.)** |
| 0.36 | 0.31 | 0.25 | 0.61 | 0.60 | 0.55 | 0.65 | 0.64 | 0.65 |
| 0.38 | 0.35 | 0.29 | 0.71 | 0.60 | 0.82 | 0.76 | 0.64 | 0.76 |
| 0.40 | 0.45 | 0.35 | 0.99 | 0.80 | 0.82 | 0.76 | 0.83 | 0.76 |

(b) $k/C$ for **COX2**.

Table 20: $k/C$ values comparing CoRGII, FHN, DiskANN-multi, and IVF-multi models under both current and pretrained GEN-embedding settings. CoRGII outperforms other methods when the impact network is not conditioned on pre-trained IsoNet embeddings.

### G.9 Estimating the error of the Chamfer approximation

Final recall computation requires impact scoring and multiprobing. It would be useful to gauge the effect of the Chamfer approximation before getting to that stage. Towards this goal, we provide the following error-based estimator to evaluate the efficacy of the Chamfer approximation without measuring recall.

We compute ground-truth distance $\mathrm{dist}^*$, distance $\mathrm{dist}_{\mathrm{soft}}$ under soft permutation, and Chamfer distance $\mathrm{dist}_{\mathrm{chamfer}}$ (see Equation 4):

$$\mathrm{dist}^* = \min_{\mathbf{P}} \sum_{i,j} \left[ \mathbf{A}_q - \mathbf{P}\,\mathbf{A}_c\,\mathbf{P}^\top \right]_+ [i,j]$$

$$\mathrm{dist}_{\mathrm{soft}} = \sum_{i,j} \left[ \mathbf{Z}_q - \mathbf{P}^{\mathrm{soft}}\mathbf{Z}_c \right]_+ [i,j]$$

The following Table 21 shows that $\mathrm{dist}^*$ is better approximated by $\mathrm{dist}_{\mathrm{chamfer}}$ than by $\mathrm{dist}_{\mathrm{soft}}$.

| Dataset | Average($|\mathrm{dist}^* - \mathrm{dist}_{\mathrm{soft}}|$) | Average($|\mathrm{dist}^* - \mathrm{dist}_{\mathrm{chamfer}}|$) |
|---|---|---|
| PTC-FR | 26.9 | **20.18** |
| PTC-FM | 35.42 | **22.53** |

Table 21: Chamfer better approximates $\mathrm{dist}^*$ (lower is better).

## H  Further discussion

In this section, we include further material of interest. Specifically, we include index memory and timing statistics for our method, a discussion on the generalizability of our impact scorer and a discussion on the sample complexity of contrastive learning.

### H.1  Index memory and timing statistics

In the Table 22 below, we report index construction time and memory usage, both of which remain modest and practical across datasets, supporting the scalability of CORGII. Further, we observe that indexing of CORGII on the PTC-FR dataset takes 46M and 272s for a **million graphs**, which is quite modest.

| Dataset | Indexing time | Index memory |
|---|---|---|
| PTC-FR | 29s | 3.6M |
| COX2 | 27.2s | 4.3M |

Table 22: Index construction time and memory usage for CORGII.

### H.2  Discussion on generalizability of impact score function

Below, we discuss the effect of node types that may occur very infrequently in the corpus, and whether there may not be sufficiently many samples to guarantee good generalization of the learned impact score function $\mathrm{Impact}_\psi$.

We emphasize that even if there are such rare graphs, we train the impact scorer using a ranking loss on the aggregated ranked list. Suppose we have relevant query–corpus pairs $G_q$ and $G_c$, each with rare tokens. During the batching procedure, our ranking loss ensures that $S(G_q, G_c) > S(G_{q'}, G_{c'})$ whenever $G_{q'}$ and $G_{c'}$ are not relevant, including for graphs that have richly contextualized tokens. This allows the training algorithm to reinforce a strong enough signal into the impact scorer. If we use $S(G_q, G_c)$ as the sole scoring function, then we obtain a modest $\mathrm{MAP} \approx 0.28$ even if $S$ is not any graph-matching model and is driven by a simple MLP.

This can be further enhanced by using the following modification of the impact scores:

$$\sum_{u \in V_q} \mathrm{Impact}_\psi\big(\hat{\mathbf{z}}_q(u), \mathbf{h}_q(u)\big)\,\mathbb{I}\big[\hat{\mathbf{z}}_q(u) \in \omega(G_c)\big] \longrightarrow \sum_{u \in V_q} \lambda\,\mathrm{Impact}_\psi\big(\hat{\mathbf{z}}_q(u), \mathbf{h}_q(u)\big)\,\mathbb{I}\big[\hat{\mathbf{z}}_q(u) \in \omega(G_c)\big]$$

Here, $\lambda$ is a hyperparameter greater than 1 and is only applied if $\hat{\mathbf{z}}_q(u)$ is a rare token.

## H.3  Discussion on sample complexity of contrastive learning

Here, we discuss the sample complexity of the contrastive learning methods that we have employed to train our pipeline (especially the impact scorer), in the context of relevant past work. Consider the resampling or re-balancing algorithm proposed in [55]. At each iteration $t$, we may use a sampling procedure, formulated as a multinomial distribution $\text{Multinomial}(D^t)$ over all corpus items (with $D^t = [D_c^t : c \in C]$) over difficulty estimates $D_c^t$ of tokenizing the graphs arising due to presence of rare tokens. Then, we re-train or fine-tune the tokenizer GTNet on graphs sampled from this distribution. In this case, we can compute predictions of tokens as $z^{(t)}(u)$ for each node $u$. Then, we leverage the prediction variations $z^{(t)}(u) - z^{(t-1)}(u)$ (and their signs) to compute difficulty for each node. Then, we can aggregate these difficulty values across nodes to compute difficulty $D_c^{(t)}$ for $G_c$. We can also compute

$$\sum_{u \in V_q} \text{Impact}_\psi\big(\hat{\mathbf{z}}_q(u), \mathbf{h}_q(u)\big)\,\mathbb{I}\big[\hat{\mathbf{z}}_q(u) \in \omega(G_c)\big] \;\longrightarrow\; \sum_{u \in V_q} \lambda\,\text{Impact}_\psi\big(\hat{\mathbf{z}}_q(u), \mathbf{h}_q(u)\big)\,\mathbb{I}\big[\hat{\mathbf{z}}_q(u) \in \omega(G_c)\big]$$

as a proxy difficulty per node. This would be a more principled alternative to our proposal in Section H.2 involving $\lambda$.

Next, the paper [56] assigns suitable weights $\alpha(x, y)$ on the loss per each instance. It would be an interesting research problem to design $\alpha(z(u), G_c)$ similar to [56]. A key challenge in immediately adapting [56] is that we may need to estimate $P(G_c \mid z(u))$ as used in Eq. (2) in [56], which would pose an interesting research question.

**Discussion of Alon et al. [57]**  Contrastive training is ubiquitous in dense retrieval, starting from such classic papers as DPR [14], ANCE [58], and RocketQA [59]. Further enhancements were reported in COIL [60], SimCSE [61], and GTR [62]. Alon et al. [57] provide extremely insightful observations on the sampling complexity of contrastive learning from a PAC point of view. They show that, for contrastive training with $n$ points in $\tilde{d}$-dimensional space, $\tilde{\Omega}(\min(nd, n^2))$ samples are needed. They provide two escape clauses, however. First, the number of samples needed reduces if the points admit some underlying clusters. Second, if each triplet comes with a demand for large separation, then the lower bound drops to $\tilde{\Omega}(n)$. In LSH-style retrieval, large separation is usually the goal, in the sense that points "near" the query should fall into the same bucket (Hamming distance 0) as the query, and "far" points (that are $\geq (1 + \epsilon)$ times farther than near points) should fall into other buckets (Hamming distance $\geq 1$). It is possible that this is what makes contrastive training work well in retrieval applications.

