# OpenReview forum: "Contextual Tokenization for Graph Inverted Indices"
_NeurIPS.cc/2025/Conference — NeurIPS 2025 poster_

### Official Review · Reviewer_69TG · 2025-06-05

**Clarity:** 2
**Significance:** 2
**Originality:** 2
**Rating:** 4
**Confidence:** 3

**Summary:**

This paper proposes a novel framework for subgraph retrieval from large graph corpora. The key idea is to discretize contextualized node embeddings (via a GNN-based tokenizer) into sparse binary tokens over a latent vocabulary, enabling efficient inverted indexing similar to classical document retrieval. CORGII further introduces trainable impact scores for tokens, replacing static metrics, and a co-occurrence-based multi-probing strategy to improve recall. Experiments across several datasets demonstrate competitive accuracy-efficiency trade-offs compared to baselines.

**Questions:**

1. What is the precise architecture of the tokenizer module? How is the vocabulary defined, and is it shared between query and corpus graphs?

2. How is the impact score computed and trained? Is it input-dependent or globally parameterized?

3. Can CORGII be used without reranking from IsoNet? What is the standalone retrieval accuracy?

4. What is the index construction time and memory footprint? How does it scale to millions of graphs?

5. How does performance change under graph structure perturbations or with unseen topologies?

**Ethical Concerns:**

["NO or VERY MINOR ethics concerns only"]

**Final Justification:**

The motivation is sound and the experimental results are great in the reported benchmarks

**Limitations:**

1. Discrete tokens may not be robust to minor structural variations in graphs.
2. Indexing large corpora with learned tokens may be memory-intensive, but this is not analyzed.
3. Heavy reliance on a pretrained IsoNet backbone suggests CORGII is not an end-to-end solution.

**Quality:**

3

**Strengths And Weaknesses:**

Strengths
1. The motivation is sound. Subgraph retrieval is a fundamental problem in applications such as molecular search and program analysis, and accelerating this process remains a relevant challenge.
2. The integration of GNNs, differentiable tokenization, and inverted indexing is a reasonable engineering direction in analogous developments in dense text retrieval.
3. Experimental results cover multiple datasets and baselines, showing great results on accuracy and latency.

Weaknesses
1. The presentation is somewhat vague in some places. The architecture and training procedure of the Graph Tokenizer Network are under-specified. It's unclear whether tokenization is learned via soft clustering, Gumbel-softmax, or discrete vector quantization. The multi-probing strategy is only described conceptually, without algorithmic detail or pseudocode. How token co-occurrence is estimated or utilized remains vague.

2. No ablation study is provided to evaluate the necessity of core components (e.g., impact score, multi-probing, GNN tokenization). Without this, it is difficult to assess what actually drives performance. Index construction cost (in terms of memory, latency, or scalability) is not discussed or measured, despite being central to the claimed efficiency.

---

> ### Author Rebuttal · Authors · 2025-07-31
>
> We would like to thank the reviewer for their comment, which will improve our paper
>
> >architecture, training  of Graph Tokenizer
>
> GTN consists of two components (1) GNN and (2) two MLPs. The pipeline begins with a shared GNN $\mathrm{GNN} _\theta$, which computes structure-aware node embeddings $\mathbf{x} _q(u)$ for query graphs  $G _q$, and $\mathbf{x} _c(v)$ for corpus graphs $G _c$. These embeddings $\mathbf{x}(u)$ are distinct from the pretrained IsoNet embeddings $\mathbf{h}(u)$.
>
> We apply separate MLPs (Eqns. 2–3) to $\mathbf{x} _q(u)$ and $\mathbf{x} _c(v)$, each ending with a sigmoid activation to produce continuous outputs in $(0,1)^D$. These yield soft tokenization vectors $\mathbf{z}(u) \in \mathbb{R}^D$ for every node. These scores are **trained using the Chamfer distance loss** (Eqn. 4) and a **ranking loss** (Eqn. 5) that encourages correct subgraph retrieval ordering.
>
> After training is completed, the continuous logits $\mathbf{z}(u) \in (0,1)^D$ are converted into binary token vectors $\widehat{\mathbf{z}}(u) \in (0,1)^D$ via hard thresholding at test time for indexing and retrieval:
> $$\widehat{\mathbf{z}}(u)[i] = \mathbf{1}[\mathbf{z}(u)[i] \ge \text{threshold}]$$
>
> Therefore, CORGII avoids sampling-based approximations like Gumbel-softmax or vector quantization and instead relies on a fully differentiable training process with late discretization.
>
> We discussed them in paper in Section 3.1. Minute details are given in Appendix D.2.
>
>
> >*Is the Vocabulary shared*
>
> The token vocabulary is implicitly defined as the set of all possible $D$-bit binary codes, i.e., $\mathcal{T} = (0,1)^D$. This vocabulary is **shared** between query and corpus graphs; however, the use of separate MLPs allows us to learn **asymmetric containment-aware representations** within a **symmetrically shared token space**—a crucial design choice for aligning with containment-style relevance while still enabling efficient token-based matching via an inverted index.
> >*Details about multiprobing*
>
> Each graph is represented as a multiset of discrete tokens aggregated from its node-level $\widehat{\mathbf{z}}(u)$ vector. In single probing, we proceed as follows: Given a token $\widehat{\mathbf{z}}(u)$, we return the graphs in the posting list corresponding to this token. In multiprobing, we also collect PostingList($\tau$), where $\tau$ is ($u$ or sufficiently) similar to $u$ in the sense that the posting lists of $\tau$ and $u$ have large overlap ― eqn. (13).
>
> CORGII compute this similarity based on **co-occurrence**. A token co-occurrence graph is constructed offline by estimating the overlap between token posting lists across the training corpus. As defined in Equation (13), a normalized similarity score is computed between tokens based on how much their posting lists overlap. At query time, for each query token $\tau _i$, we retrieve both its posting list and those of its top-$k$ co-occurring tokens $\tau _j$.
> > *How is impact score computed and trained? is it input dependent/globally paremeterized*
>
> $\text{Impact} _\psi(\tau _u, \text{Embedding}(u))$ is an MLP which takes the binary embedding of $\tau$, i.e., $\hat{z}$ and some continuous embedding (intermediate GNN embeddings or pretrained embeddings) as input and output a scalar value. This design makes the impact score **input-dependent** and **query-aware**, rather than being globally parameterized or fixed like traditional TF-IDF or BM25 weights. This is trained by minimizing a pairwise ranking loss (Eqn. 10).
> >Ablation study.
>
> We presented several ablation study in the experiments (see L343 to L383). We did not clearly highlight them as ablation study and we are sorry for that. We will mention them in the revised version.
>
> First we perform ablation on multiprobing, by comparing between co-occurence based multi-probing (CM), Hamming multiprobing (HM in Eq. 12) and single probing. Following table shows the results for PTC-FR which shows that CORGII performs best and Single probe is worst.
>
> |$k/C$|CORGII (CM)|HM|Single Probe|
> |-|-|-|-|
> |0.1|0.17|0.16|0.16|
> |0.2|0.27|0.25|0.25|
> |0.3|0.34|0.32|0.31|
> |0.4|0.38|0.36|0.36|
> |0.5|0.42|0.4|0.38|
> |0.6|0.44|0.42|0.38|
> |0.7|0.46|0.44|0.38|
> |0.8|0.47|0.48|0.38|
> |0.9|0.48|0.48|0.38|
> |1.0|0.48|0.48|0.48|
>
> Next, we perform ablation on impact scores, by comparing our method against uniform impact (equal impact for all tokens and all corpus graphs). Following table shows the results for PTC-FR which shows that CORGII performs better than uniform impact.
>
> |$k/C$|CORGII|Unif. impact|
> |-|-|-|
> |0.1|0.17|0.09|
> |0.2|0.27|0.17|
> |0.3|0.34|0.23|
> |0.4|0.38|0.28|
> |0.5|0.42|0.33|
> |0.6|0.44|0.37|
> |0.7|0.46|0.40|
> |0.8|0.47|0.43|
> |0.9|0.48|0.46|
> |1.0|0.48|0.48|
>
> Next, we use a Siamese module, instead of our current tokenizer module. The following table shows the results for PTC-FR, which shows that CORGII performs better than the alternative.
>
> |$k/C$ | CORGII | Siamese|
> |-|-|-|
> |0.1|0.17|0.03|
> |0.2|0.27|0.07|
> |0.3|0.34|0.11|
> |0.4|0.38|0.16|
> |0.5|0.42|0.20|
> |0.6|0.44|0.26|
> |0.7|0.46|0.30|
> |0.8|0.47|0.36|
> |0.9|0.48|0.42|
> |1.0|0.48|0.48|
> >Index construction cost ...Scaling for million graphs.
>
> We observe that indexing of CORGII on the PTC-FR dataset takes 46MB and 272s for a million graphs, which is quite modest. Next we provide a theoretical complexity for time and space.
>
> Suppose the vocabulary size is $M$. Posting list lengths typically follow Zipf's law. Hence, by definition of Zipf law, the probability distribution of occurrence of the $k$th most frequent token (in the corpus) is $f(k) = \frac{A}{k^{\delta}}$ with $0.5<\delta < 1$.
> $$\int  _{1} ^M f(k)dk = 1 \implies A\frac{M^{1-\delta}}{(1-\delta)} - \frac{A}{1-\delta}=1$$
> Since $M$ is large, we have $$A = O(M^{\delta-1})$$
>
> Let the aggregated length of posting lists is $L$. Then, assuming the $M$-th popular token will occur only once, we have:
> $$\frac{LA}{k^{\delta}} = 1 \text{ at } k = M$$
> This implies $L = O(M)$. For one query token, the expected number of corpus graphs returned will be the expected length of the posting list of the token. The expected query cost is thus $|V _q| \mathbb{E} (\text{Length of posting list})$, where
> $$\mathbb{E} (\text{Length of posting list}) = \int  _{1} ^{M} Lf(k) \times f(k) dk = O(M M^{2\delta-2})  \int  _{1} ^{M} {k^{2\delta}} dk$$
> Now, $\int  _{1} ^{M} {k^{2\delta}} dk =$ constant since $\delta\in (0.5,1)$. Hence, the expected time complexity of our method is $O(M^{2\delta-1})$. Similarly, the space complexity of index is the total length of posting list $O(L) = O(M)$.
>
> On the other hand, for the single vector representation in FHN, the seminal paper of (Indyk and Motwani, 1998) showed that the time complexity is $O(|C|^{\gamma})$ where $\gamma<1$ is the efficiency parameter of LSH. The space complexity is $O(|C|^{1+\gamma})+O(|C|)$.
> Since in practice, number of corpus items is much larger than size of the vocabulary, i.e., $|C| \gg M$, CORGII is more efficient than single vector retrieval in terms of time and accuracy (also see following table).
>
> |Running time for **PTC-FR**|CoRGII|FHN|
> |-|-|-|
> |MAP = 0.38|0.04s|0.06s|
> |MAP = 0.45|0.08s|0.12s|
>
> > Can CORGII be used without reranking from IsoNet?
>
> Many contemporary search systems are built as a cascade of a *retriever*  and a *reranker*. CORGII is fundamentally a *retriever*, which, given a query, effciently cuts down the corpus to a much smaller *subset* of eligible candidates. How this subset is scored and reranked is orthogonal to the retriever.
>
> Hence, CORGII can be used standalone to return a set of IDs, just like any other retriever like DiskANN, LSH etc. It does not rank the documents. Hence, like any other retriever, we must need some ranker to rank those documents. But, CORGII is not dependent on fixed reranking policy. It can be paired with any ranking policy.
>
> We used IsoNet because it is a SOTA reranker. Instead of this, if we use some other reranker like GEN [A], we observed significant degradation of the accuracy-speed trade off of the baselines (MAP from 0.42 to 0.37 for k/C = 0.7; MAP  from 0.38 to 0.33 for k/C = 0.5), while we still perform better.
>
> [A] Graph Matching Networks for Learning the Similarity of Graph Structured Objects. Li et al, ICML 2019.
>
> > Heavy reliance on a pretrained IsoNet backbone .
>
> Note that we did not pre-train our GNN model (Fig 1). Hence, the tokenizer operates without any pre-trained GNN and therefore, the resulting index ― built solely on the tokens ― is also completely independent of pre-trained embeddings.
>
> Only our impact scoring model does utilize pre-trained embeddings from IsoNet. However, this results in minor dependence of pre-trained embeddings. We illusrate this by using two alternatives of current pre-trained embeddings from IsoNet
>
> (1) Instead of pre-trained embeddings $h$,  we use $x$ to as input to the impact scoring model (Impact($\tau,x$), Eq. 8).
>
> (2) Instead of training $h$ on the same dataset, we train $h$ on PTC-FM dataset and apply it to PTC-FR.
>
> Following tables shows the efficiency in terms of fraction of graphs $k/C$ retrieved to achieve atleast a certain MAP=$m^*$ for PTC-FR   datasets. Lower k/C is better and indicates higher efficiency.
>
> |$m^*$|Our (current)|Our (use $x$)|Our ($h$ trained on other dataset)|FHN|
> |-|-|-|-|-|
> |0.38|0.38|0.376|0.42|0.51|
> |0.40|0.44|0.436|0.47|0.81|
> |0.45|0.645|0.626|0.69|0.99|
>
> We observe that the efficiency does not change much and even if we do not use pre trained embeddings, our method outperforms baselines.
>
> > Performance change under graph structure perturbations
>
> We perform experiments with noised PTC-FR graphs, which results in $\sim 10\\%$ noise in embedding space.
> We report the MAP values  for different values of k/C (retrieved graph fraction for PTC-FR), which shows that our method performs better.
> |k/C|CoRGII|FHN|
> |-|-|-|
> |0.1|0.18|0.15|
> |0.2|0.28|0.25|
> |0.3|0.35|0.34|
> |0.4|0.38|0.34|
> |0.5|0.42|0.38|
> |0.6|0.44|0.38|
> |0.7|0.46|0.42|
> |0.8|0.47|0.42|
> |0.9|0.48|0.48|
> |1.0|0.48|0.48|

---

> ### Author Response · Authors · 2025-08-04
> **Request to reply**
>
> Dear reviewer 69TG,
>
> We addressed your comments, to the best of our abilities. Could you please let us know if you have any concerns/questions? Did our rebuttal address your concerns?

---

> > ### Comment · Reviewer_69TG · 2025-08-06
> >
> > Thanks for the response and I will change the score.

---

### Official Review · Reviewer_DMJr · 2025-07-02

**Clarity:** 3
**Significance:** 3
**Originality:** 3
**Rating:** 4
**Confidence:** 3

**Summary:**

This paper introduces CORGII, a novel framework for the retrieval of graphs containing a given query subgraph. The core idea is to represent each graph as a set of discrete, structure-aware tokens generated by GNN and a differentiable binarization process. These tokens make it possible to leverage classic inverted index techniques (which were originally developed for text retrieval) to index and search large graph collections efficiently. Furthermore, the use of co-occurrence-based multi-probing enhances recall by expanding the set of candidate matches. In summary, CORGII effectively combines the expressive power of dense neural graph representations with the scalability and efficiency of sparse inverted indices, enabling practical and high-performance subgraph retrieval.

**Questions:**

1.	In the paper you mention, “as a first step toward tokenization, we apply two (still differentiable, but distinct) networks to hq(u), hc(v).” According to Figure 2, are these corresponding to x_q(u) and x_c(v)? There seems to be some confusion or mixing between x(u) and h(u) in the text and diagrams.
2.	In Section 3.2 (“Discretization and inverted index”), you mention binarizing z by thresholding at 0.5. Have you considered using more than one bit per element of z (i.e., multi-level or quantized discretization) to increase accuracy? What would be the trade-offs in terms of retrieval accuracy, index size, and efficiency? A brief discussion or experiment on this design choice would be helpful.
3.	You note that “if multiple nodes u have the same token \hat{z}_q(u), they are counted multiple times.” I am concerned this might violate the injectivity constraint in subgraph isomorphism/graph matching. How does this affect retrieval quality or correctness? Would deduplication or enforcing uniqueness help, or is there a principled reason for allowing duplicates?
4.	The current evaluation focuses on MAP versus retrieval set size, but does not report practical efficiency metrics such as query-per-second (QPS) versus recall (as is common in ANN literature). Could you provide this type of result?

**Ethical Concerns:**

["NO or VERY MINOR ethics concerns only"]

**Final Justification:**

Most of my concerns have been adressed. I will increase my score.

**Limitations:**

yes

**Quality:**

3

**Strengths And Weaknesses:**

Strengths:

1.	Inspired by text retrieval, CORGII combines neural graph embeddings with inverted index structures to enable efficient subgraph retrieval.

2.	The method is tested on several benchmark datasets and compared with existing baselines.

3.	The paper is well-motivated, clearly explains each system component, and includes ablation studies to show the impact of its design choices.


Weaknesses:

1.	The presentation could be improved. Specifically, the framework would benefit from a clearer breakdown of each component; In addition, the distinction between variables such as x(u) and h(u) could be clarified.
2.	The paper does not provide in-depth theoretical analysis or guarantees regarding the effects of tokenization or multi-probing on retrieval performance.
3.	The evaluation mainly reports MAP versus retrieval set size, but does not present retrieval time or runtime efficiency results, making it less convincing for readers concerned with practical scalability.

---

> ### Author Rebuttal · Authors · 2025-07-31
>
> We would like to thank the reviewer for their comment, which will improve our paper.
>
>
> > The presentation could be improved. Specifically, the framework would benefit from a clearer breakdown of each component; In addition, the distinction between variables such as x(u) and h(u) could be clarified.
>
> We agree that clearly distinguishing between $\mathbf{x}(u)$ and $\mathbf{h}(u)$ is essential. To clarify:
>
> - $\mathbf{h}(u)$ denotes **contextual node embeddings from a pretrained subgraph matching model** (IsoNet). These are **used exclusively as input to the impact scoring network**, both during training and inference. They do **not** influence the tokenization process. In other words, our GNN is trained without any pretraining and as a result, the subsequent tokenization and **indexing does not depend on pre-trained embeddings.**
>
> - $\mathbf{x}(u)$ refers to the contextual node representations **computed within CORGII itself**, using a shared GNN applied to both query and corpus graphs. These representations are input to **separate MLPs** that generate tokenization logits $\mathbf{z}(u) \in (0, 1)^D$. After training, these logits are thresholded to obtain binary codes $\widehat{\mathbf{z}}(u) \in (0,1)^D$, which are then mapped to token IDs for inverted indexing.
>
> Importantly, the **tokenization and retrieval pipeline relies solely on $\mathbf{x}_q(u)$**, as shown in **Figure 1**. The pretrained embeddings $\mathbf{h}(u)$ from IsoNet are **only used as auxiliary inputs for impact scoring**, and are not involved in computing or indexing token representations.  A different source of node encoding than IsoNet can readily be plugged in, replacing IsoNet.
>
> There is a typo in the text: It should be written as "we apply two networks to xq(u), xc(v)" and   Figure 1is correct. Many thanks for pointing this out.
>
>
>
>
> > The paper does not provide in-depth theoretical analysis or guarantees regarding the effects of tokenization or multi-probing on retrieval performance.
>
>
> Here is some basic analysis of the complexity of CORGII under certain common assumptions on the corpus.  Suppose the vocabulary size is $M$. Posting list lengths typically follow Zipf's law, as we see in the literature [A] and also shown in our problem in App F.4 (page 25, Appendix). By Zipf's law, the probability distribution of occurrence of the $k$th most frequent token (in the corpus) is $f(k) = \frac{A}{k^{\delta}}$ with $0.5<\delta < 1$.
>
> $$\int _{1} ^M f(k)dk = 1 \implies A\frac{M^{1-\delta}}{(1-\delta)} - \frac{A}{1-\delta}=1$$
>
> Since $M$ is large, we have $$A = O(M^{\delta-1})$$
>
> Let the aggregated length of posting lists is $L$. Then, assuming the $M$-th popular token will occur only once, we have:
> $$ \frac{LA}{k^{\delta}} = 1 \text{ at } k = M$$
>
> This implies $L = O(M)$. For one query token, the expected number of corpus graphs returned will be the expected length of the posting list of the token.  The expected query cost is thus $|V _q| \mathbb{E} (\text{Length of posting list})$, where
>
> $$\mathbb{E} (\text{length of posting list}) \approx \int _{1} ^{M} Lf(k) \times f(k) dk = O(M M^{2\delta-2})  \int _{1} ^{M} {k^{2\delta}} dk $$
>
> Now, $\int _{1} ^{M} {k^{2\delta}} dk =$ constant since $\delta\in (0.5,1)$. Therefore, the expected posting list length is $O(M^{2\delta-1}) \in [O(\sqrt{M}),O(M)].$ Hence, the expected time complexity of our method
> is $O(M^{2\delta-1})$. Similarly, the space complexity of index is the total length of posting list $O(L) = O(M)$
>
>
> On the other hand, for the single vector representation in FHN, the seminal paper of (Indyk and Motwani, 1998) showed that the time complexity is $O(|C|^{\gamma})$ where $\gamma<1$ is the efficiency parameter of LSH. The space complexity is $O(|C|^{1+\gamma})+O(|C|)$.
>
> Since in practice, number of corpus items is much more larger than size of the vocabulary, i.e., $|C| \gg M$, our method is more efficient than single vector retrieval in terms of time and accuracy. Moreover, this is consistent with text based retrieval where inverted index is widely adopted over single vector based indexing.
>
> Extending to multi-token queries is messy, because then we need to model a _joint_ distribution over words in a very high dimensional space. For this reason, there has been no serious analysis of this form in the Information Retrieval literature.
>
>
> > The evaluation mainly reports MAP versus retrieval set size, but does not present retrieval time or runtime efficiency results, making it less convincing for readers concerned with practical scalability.
>
> We present a modified view of our results which directly addresses retrieval latency and scalability.  The following table reports both the size of the candidate pool and the corresponding retrieval latency (in seconds). As seen, CORGII outperforms all the baselines in terms of candidate efficiency and query time at comparable MAP thresholds (m*). CORGII retrieves significantly fewer candidates while achieving competitive or better latency and MAP, clearly demonstrating its practical scalability and efficiency.
>
>
> #### **PTC-FR**
>
> | MAP ≥ m* | FHN         | CoRGII        | IVF | DiskANN|
> |-----------------|-------------|---------------|-----------|----------------|
> | 0.38            | 0.06s | 0.04s   | 0.09s    | 0.10s         |
> | 0.40            | 0.10s | 0.05s   | 0.10s    | 0.10s        |
> | 0.45            | 0.12s | 0.08s | 0.11s     | 0.11s         |
>
> #### **COX2**
>
> | MAP ≥ m* | FHN         | CoRGII        | IVF | DiskANN |
> |-----------------|-------------|---------------|-----------|----------------|
> | 0.36            | 0.08s | 0.03s   | 0.07s     | 0.06s         |
> | 0.38            | 0.09s | 0.04s   | 0.07s    | 0.9s       |
> | 0.42            | 0.12s | 0.05s   | 0.9s     | 0.09s         |
>
> > In the paper you mention, “as a first step toward tokenization, we apply two (still differentiable, but distinct) networks to hq(u), hc(v).” According to Figure 2, are these corresponding to x_q(u) and x_c(v)? There seems to be some confusion or mixing between x(u) and h(u) in the text and diagrams.
>
> Thank you for pointing this out. You are absolutely right — the line in question should refer to $x_q(u)$ and $x_c(v)$, not $h_q(u)$ and $h_c(v)$. As clarified in our earlier response, CORGII uses GNN-derived contextual node embeddings x(u) for tokenization and impact scoring. The pretrained embeddings $h(u)$ are used only during training to supervise the impact scoring function and are never passed as inputs to any component of the CORGII architecture.
>
> We will correct this line in the final version to avoid confusion and ensure consistency with Figure 1 and the overall description of the model.
>
>
> > What would be the trade-offs in terms of retrieval accuracy, index size, and efficiency.
>
> We change the discretization, which resulted in different posting list sizes. Following is the table, which shows that with respect to posting list, MAP is robust.
>
>
>
> |Average Posting List Length|MAP|
> |-|-|
> |1642.9|0.31|
> |1792.28|0.31|
> |965.4|0.29|
>
>
> Indexing of CORGII on the PTC-FR dataset takes 46MB and 272s for a million graphs, which is quite modest.
>
> We also compare CoRGII against FHN, showing that CoRGII achieves the same MAP up to 2× faster than FHN on PTC-FR dataset:
>
> |$m^*$|FHN| CoRGII|
> |-|-|-|
> |0.38|0.06s|0.04s|
> |0.40|0.10s|0.05s|
> |0.45|0.12s|0.08s|
>
> > You note that “if multiple nodes u have the same token $\hat{z}_q(u)$, they are counted multiple times.” I am concerned this might violate the injectivity constraint in subgraph isomorphism/graph matching. How does this affect retrieval quality or correctness? Would deduplication or enforcing uniqueness help, or is there a principled reason for allowing duplicates?
>
>
> We consider multiset-based token matching is more faithful to subgraph matching than enforcing set-based uniqueness. In subgraph isomorphism, it is possible for multiple nodes to map to the same structure or token, and discarding duplicates would remove crucial count-based information. By retaining token multiplicity, CORGII preserves these mass constraints, enabling more accurate retrieval aligned with the structural semantics of subgraph matching.
>
> Below, we compare retrieval performance with and without deduplication for two datasets **PTC-FR** (first table) and **COX2** (second table). We observe that deduplication **slightly reduces recall**, confirming that duplicate tokens improve retrieval quality in our setup.
>
>  **PTC-FR**
> |$k/C$|Original|Dedup|
> |---|--------|-----|
> |0.1|0.17|0.15|
> |0.2|0.27|0.25|
> |0.3|0.34|0.32|
> |0.4|0.38|0.36|
> |0.5|0.42|0.40|
> |0.6|0.44|0.41|
> |0.7|0.46|0.43|
> |0.8|0.47|0.44|
> |0.9|0.48|0.46|
> |1.0|0.48|0.48|
>
> **COX2**
> |$k/C$|Original|Dedup|
> |---|--------|-----|
> |0.1|0.22|0.17|
> |0.2|0.32|0.28|
> |0.3|0.39|0.35|
> |0.4|0.44|0.40|
> |0.5|0.48|0.45|
> |0.6|0.51|0.48|
> |0.7|0.53|0.50|
> |0.8|0.55|0.52|
> |0.9|0.56|0.54|
> |1.0|0.56|0.56|
>
>
>
> > The current evaluation focuses on MAP versus retrieval set size, but does not report practical efficiency metrics such as query-per-second (QPS) versus recall (as is common in ANN literature). Could you provide this type of result?
>
>
>
> |Throughput on PTC-FR Dataset|Queries|Corpus Items|Query-per-second|
> |-------------|-------|------------|----------------|
> |CoRGII|100|38K|555.55|
> |FHN|75|51K|357.14|
>
>
>
> |Throughput on COX2 Dataset|Queries|Corpus Items|Query-per-second|
> |-------------|-------|------------|----------------|
> |CoRGII|123|31K|1021.75|
> |FHN|63|61K|311.6|
>
> We observe that our QPS is much higher.

---

> > ### Author Response · Authors · 2025-08-06
> >
> > Dear Reviewer DMJr,
> >
> >
> > We would greatly appreciate it if you could let us know whether our response has addressed your concerns. If there are any remaining issues or if further clarification would be helpful, we would be more than happy to provide additional details.
> >
> >
> >
> > Regards,
> >
> > Authors

---

> > > ### Comment · Reviewer_DMJr · 2025-08-06
> > >
> > > Thanks for the detailed responses. I would like to increase my rating.

---

> ### Author Response · Authors · 2025-08-05
> **Your question  on "more than one bit per element of z " to increase accuracy**
>
> Dear reviewer DMJr,
>
> First, we would like to thank you for your review. We found it really insightful.
>
> Lately, we realized that we could have better answered your question on multi-level quantization
>
> > *Have you considered using more than one bit per element of z (i.e., multi-level or quantized discretization) to increase accuracy?*
>
> We performed experiments and observed the following results.
>
> |$k/C$|Single-Probe(0 bit)|1 bit|3 bits|7 bits|
> |---|------------------|------|------|------|
> |0.1|0.16|0.16|0.16|0.12|
> |0.2|0.25|0.25|0.25|0.21|
> |0.3|0.31|0.31|0.32|0.29|
> |0.4|0.36|0.36|0.36|0.25|
> |0.5|0.38|0.39|0.40|0.39|
> |0.6|0.38|0.41|0.42|0.43|
> |0.7|0.38|0.41|0.44|0.45|
> |0.8|0.38|0.41|0.48|0.46|
> |0.9|0.38|0.41|0.48|0.48|
> |1.0|0.48|0.48|0.48|0.48|
>
> We observe that multiple bits improve the performance.  There results indicate that there is a sweet spot around 3 bits, which provides very good tradeoff. **In Appendix F.1, we already provided results in a figure format in more detail.**

---

### Official Review · Reviewer_R7bo · 2025-07-06

**Clarity:** 2
**Significance:** 3
**Originality:** 4
**Rating:** 5
**Confidence:** 4

**Summary:**

The authors propose CORGII, a method for graph retrieval. Given a small query graph, and a corpus consisting of a collection of several graphs, the goal is to retrieve a list of corpus graphs that contain the query graph. In principle, this requires solving an NP-hard optimization problem, which motivates their method that relaxes the problem into approximate search using vector representations of the query and corpus graphs. CORGII uses GNNs to compute binary representations of nodes in query and corpus graphs, which can in turn be used to compute scores of candidate graphs in an efficient way. The GNNs in CORGII are trained with an approximate objective that seeks to match embeddings of query nodes and notdes in relevant graphs in the corpus. The authors additionally propose several ways to compute scores to account for the information lost in binary representations.

**Questions:**

1. What is the effect of the specific data used to train IsoNet on the performance of CORGII? How well do you expect it to generalize?
2. Could you please clarify whether IsoNet is used to rerank all baselines during the evaluation, and if so, whether the comparison is fair since CORGII is trained with IsoNet embeddings?
3. What is the cost of running CORGII in comparison with FHN?
4. Did you consider approaches like continuous relaxation of discrete variables when designing GTNet?

- **Clarification questions:**

  4. In Line 22: "$K$ graphs that 'best' contain $G_c$"---shouldn't this be $G_q$?

  5. L105: What is the intepretation of $\leq$ in this context?

  6. L129: The posting list is first defined as a function of $\tau$ (PostingList($\tau$)), and later as a function of $c$ (PostingList($c$)). Which is the right one?

**Ethical Concerns:**

["NO or VERY MINOR ethics concerns only"]

**Final Justification:**

CORGII is an efficient and performant method for graph indexing. The evaluation methodology and empirical results are strong, and the dicussion has led to additional insights that point to its effectiveness, thus I have decided to increase my score. The work can still be improved in terms of clarity, which I encourage the authors to address.

**Limitations:**

The authors do discuss some limitations in the Appendix, though I would suggest to indicate the most critical ones in the main body of the paper to better inform readers of the feasibility and implications of the proposed method.

**Paper Formatting Concerns:**

No concerns.

**Quality:**

3

**Strengths And Weaknesses:**

**Strengths**

1. The problem of graph retrieval is well-motivated and the proposed approach is novel. Different model decisions, such as using different MLPs in the GTNet, the use of the Chamfer distance, the impact networks, and multi-probing strategies are well motivated in relation to issues that are likely to occur during retrieval.
2. The work seems to be well-positioned with respect to the related work.
3. The experiments indicate that CORGII outperforms the baselines in the task, over a broad budget (i.e. fraction of the corpus retrieved) range.
4. The authors provide additional ablation experiments that for the most part support the efficacy of the different parts in CORGII.

**Weaknesses**

1. The method relies on a pre-trained GNN to compute node representations. This could potentially limit how well CORGII generalizes, for example to graphs very different to those used to train IsoNet. This is an important part of CORGII that in my opinion should be discussed in the main body.
2. One aspect not entirely clear to me during the evaluation is the reranking. Sec. 4, L307 states: *"For each query* $\mathcal{Q}_{\text{test}}$*, we retrieve the corpus graphs* $\mathcal{R}_q$ *that are relevant by the corresponding model. We rerank the retrieved candidates using the pretrained ranking model.*" Does this mean that CORGII scores, as well as the baselines, are reranked by IsoNet? If this is the case, the comparison is not entirely fair because CORGII is trained with IsoNet embeddings.
3. Interestingly, FHN seems to closely follow CORGII in performance, all while being a much simpler and possible cheaper baseline that averages node embeddings. A discussion on the cost of CORGII in comparison with such a simple baseline would help better contextualize the observed benefits.
4. The performance of FHN also leads to some dubious claims, such as there being a "benefit of retaining node-level granularity" (L331). That FHN, a single-vector method, outperforms IVF-multi and DiskANN-multi suggests otherwise.
5. There are several works on learning discrete representations in neural networks, which appears to not have been considered. GTNet is trained to compute continuous representations, but at test time these are discretized, which causes a mismatch between the training and test distribution. An alternative is to rely on relaxed distributions which try to approximate discrete variables while still allowing gradient-based learning, see e.g. [1].
6. There are some issues with clarity in the paper, see questions.

Overall, I believe this is a technically solid paper, and I would be open to raising my score if the authors are able to address these weaknesses and my questions.

**References**

[1] Madddison et al. (ICLR 2017), "The Concrete Distribution: A Continuous Relaxation of Discrete Random Variables".

---

> ### Author Rebuttal · Authors · 2025-07-31
>
> We would like to thank the reviewer for their comment, which will improve our paper
>
>
> > *The method relies on a pre-trained GNN.... the effect of the specific data used to train IsoNet*
>
>
> We clarify that our GNN model is not pre-trained; it is trained independently for indexing.  Figure 1 shows the correct picture. $\text{GNN}_\theta$ is not informed by and has no connection to the pretrained node embeddings $h$. Moreover, we also clarified in lines 165–167 and the loss function clearly shows that GNN model parameter theta is trained using Chamfer scores and it is not pretrained.
>
>
> Since GNN does not use pretrained embeddings,  tokenization network  also does not rely on any pre-trained embeddings. Hence, the resulting index ― built solely on the tokens ― is also completely of pre-trained embeddings.
>
>
> However, our impact scoring model does utilize pre-trained embeddings $h$ from IsoNet. This results in minor dependence of pre-trained embeddings, which does not affect our performance. IsoNet can be readily replaced by any other scoring method. Scoring methods can even transfer between domains.
>
>
> To demonstrate this, instead of pre-training $h$ on the same dataset, we train h on the PTC-FM dataset and apply it to PTC-FR, and likewise train on PTC-MR and apply to COX2. Following tables show  the efficiency in terms of fraction of graphs $k/C$ retrieved to achieve atleast a certain MAP=$m^*$. Lower k/C is better and indicates higher efficiency.
>   _ denotes places where the method is unable to achieve target MAP.
>
> **k/C values for PTC-FR**
>
> |$m^*$|Our (current) |Our ($h$ trained on PTC-FM dataset)|FHN|FHN (uses $h$ trained on PTC-FM dataset) |IVF|IVF (uses $h$ trained on PTC-FM dataset)
> |-|-|-|-|-|-|-|
> |0.38|0.38|0.42|0.51|0.97|0.76|0.81|
> |0.40|0.44|0.47|0.81|1.0|0.89|0.81|
> |0.45|0.645|0.69|0.99|1.0|0.96|0.91|
>
>
>
> **k/C values for COX2**
> |$m^*$|Our (current) |Our ($h$ trained on PTC-MR dataset)|FHN|FHN (uses $h$ trained on PTC-MR  dataset) |IVF|IVF (uses $h$ trained on PTC-MR  dataset)
> |-|-|-|-|-|-|-|
> |0.36|0.31|0.27|0.61|1.0|0.64|0.57|
> |0.38|0.35|0.29|0.71|-|0.64|0.81|
> |0.40|0.45|0.37|0.99|-|0.83|0.81|
>
> Even if we use IsoNet embeddings trained on other datasets, CORGII beats baselines. Moreover, it is more robust with respect to changing those embeddings.
>
>
> > *CORGII scores, as well as the baselines, are reranked by IsoNet?*
>
>
> **Use of IsoNet reranker:** Many contemporary search systems are built as a cascade of a *retriever* and a *reranker*.  CORGII is fundamentally a *retriever*, which, given a query, efficiently cuts down the corpus to a much smaller *subset* of eligible candidates. How this subset is scored and reranked is orthogonal to the retriever.
>
> It is essential that competing retrievers are compared keeping the reranker fixed. Given that IsoNet offered a competitive scoring function, we used IsoNet as reranker. This gives all baselines the same benefit. If we use different rerankers, the difference in performance is difficult to "factor" across retriever and reranker.
>
> **On fair comparison:** All baselines use pretrained embeddings. FHN uses Fourier features from the pretrained embeddings and train an intermediate embeddings based on those pretrained embeddings based Fourier features. Since the baselines require pre-trained embedding vectors, we use the same pre-trained embeddings across all baselines including our (in impact scoring model). Hence, our comparison is fair ― we do not use any additional signals as input, that the baselines cannot access.
>
> In CORGII, the **GNN is not pretrained and does not use any pretrained embeddings**.  Pretrained embeddings $h$ are used only to learn impact scores. **Any** pretrained embedding leading to scores can be used in place of IsoNet here.
>
> Through two experiments, we further show that the superiority of our method holds true even if we use different embeddings.
>
> (1) First, instead of pre-trained embeddings $h$,  we use $x$ to as input to the impact scoring model (Impact($\tau,x$), Eq. 8), where $x$ is the embedding from the CorGII GNN, (which is not pretrained and trained as a part of CorGII scheme).
>
> Following tables show  the efficiency in terms of fraction of graphs $k/C$ retrieved to achieve atleast a certain MAP=$m^*$ for **PTC-FR (first table) and COX2 (second table) datasets**. Lower k/C is better and indicates higher efficiency.
>
> **k/C for PTC-FR**
> |$m^*$|Our (current) | Our (use $x$)|FHN|DiskANN-multi|IVF-multi|
> |-|-|-|-|-|-|
> |0.38|0.38|0.37|0.51|0.87|0.76|
> |0.4|0.44|0.43|0.81|0.87|0.89|
> |0.45|0.64|0.62|0.99|0.93|0.96|
>
> **k/C for COX2**
> |$m^*$|Our (current) | Our (use $x$)|FHN|DiskANN-multi|IVF-multi|
> |-|-|-|-|-|-|
> |0.36|0.31|0.42|0.61|0.55|0.64|
> |0.38|0.35|0.48|0.71|0.82|0.64|
> |0.42|0.45|0.58|0.99|0.82|0.83|
>
> We observe that our performance is better, regardless of using the nature of embeddings.
>
> (2) We pre-train a different model called Graph Embedding Network (GEN)[A], which is different from the final reranking model and use them for all methods and our (again, only impact scoring). Following are the results, with _ denoting places where the method is unable to achieve target MAP:
>
>
> **k/C for PTC-FR**
>
> |$m^*$|Our (current) | Our (pretrained GEN-embedding)|FHN (current)| FHN(pretrained GEN-embedding)|DiskANN-multi|DiskANN-multi (pretrained GEN-embedding)|IVF-multi|IVF-multi (pretrained GEN-embedding)|
> |-|-|-|-|-|-|-|-|-|
> |0.38|0.38|0.38|0.51|1.0|0.87|0.82|0.76|0.82|
> |0.40|0.44|0.44|0.81|-|0.87|0.88|0.89|0.94|
> |0.45|0.64|0.65|0.99|-|0.93|0.98|0.96|0.94|
>
>
> **k/C for COX2**
>
> |$m^*$|Our (current) | Our (pretrained GEN-embedding)|FHN (current)| FHN(pretrained GEN-embedding)|DiskANN-multi|DiskANN-multi (pretrained GEN-embedding)|IVF-multi|IVF-multi (pretrained GEN-embedding)|
> |-|-|-|-|-|-|-|-|-|
> |0.36|0.31|0.25|0.61|0.6|0.55|0.65|0.64|0.65|
> |0.38|0.35|0.29|0.71|0.6|0.82|0.76|0.64|0.76|
> |0.40|0.45|0.35|0.99|0.8|0.82|0.76|0.83|0.76|
>
>
>
>
> We observe that our performance is better, regardless of using the nature of embeddings.
>
>
>
>
> [A] Graph Matching Networks for Learning the Similarity of Graph Structured Objects. Li et al, ICML 2019.
>
>
> > *cost of CORGII in comparison with FHN*
>
> FHN is not that cheaper. It samples frequencies, feeds the resulting Fourier vectors into a neural network to learn a Fourier map, and then uses this map in a neural hashcode generator.
> In contrast, CORGII discretizes dense graph embeddings into tokens, enabling fast retrieval using inverted indices—similar to text search. Unlike dense retrieval with single or multiple probes, CORGII operates with discrete tokens, making it faster. Next, we give a complexity analysis
>
>
>
> Suppose the vocabulary size is $M$. Posting list lengths typically follow Zipf's law. Hence, by definition of Zipf law, the probability distribution of occurrence of the $k$th most frequent token (in the corpus) is $f(k) = \frac{A}{k^{\delta}}$ with $0.5<\delta < 1$.
>
>
> $$\int _{1} ^M  f(k)dk = 1 \implies A\frac{M^{1-\delta}}{(1-\delta)} - \frac{A}{1-\delta}=1$$
>
> Since $M$ is large, we have $$A = O(M^{\delta-1})$$
>
>
> Let the aggregated length of posting lists is $L$. Then, assuming the $M$-th popular token will occur only once, we have:
>
> $$ \frac{LA}{k^{\delta}} = 1 \text{ at } k = M$$
>
> This implies $L = O(M)$. The expected number of retrieved graphs per token will be expected length of posting list under Zipf law. Hence, the expected number of retrived graphs is $|V_q| \mathbb{E} (\text{Length of posting list})$, where
>
> $$\mathbb{E} (\text{Length of posting list}) = \int _{1} ^{M} Lf(k) \times f(k) dk = O(M M^{2\delta-2})  \int _{1} ^{M} {k^{2\delta}} dk $$
>
> Now, $\int _{1} ^{M} {k^{2\delta}} dk =$ constant since $\delta\in (0.5,1)$.  Hence, the expected time complexity of our method
> is $O(M^{2\delta-1})$. Similarly, the space complexity of index is the total length of posting list $O(L) = O(M)$
>
>
> Indyk and Motwani (1998) showed that the time complexity FHN is $O(|C|^{\gamma})$ where $\gamma<1$ is the efficiency parameter of LSH. The space complexity is $O(|C|^{1+\gamma})+O(|C|)$.
>
> In practice, number of corpus items is much more larger than size of the vocabulary, i.e., $|C| \gg M$, CORGII is more efficient than single vector retrieval in terms of time and accuracy.
> CORGII is also fast in practice.
>
> |Running time for **PTC-FR dataset**|CoRGII|FHN|
> |-|-|-|
> |MAP = 0.38|0.04s|0.06s|
> |MAP = 0.45|0.08s|0.12s|
>
>
> > *FHN, a single-vector method, outperforms IVF-multi and DiskANN-multi*
>
> IVF-multi and DiskANN-multi indices are built using norm-based distances (e.g., Euclidean) between query and corpus embeddings. In contrast, our subgraph matching task requires ranking based on hinge distance between query and corpus embeddings. Incorporating hinge distance into these methods is non-trivial, as their indexing relies on corpus–corpus distances rather than query–corpus distances.
>
> On the other hand, FHN is specifically designed for hinge distance. Therefore, despite single vector based representation, it performs better.
>
>
> > *An alternative is to rely on relaxed distributions. [1].*
>
>
> We experimented with the approach in [1] by incorporating a relaxed, continuous approximation to the tokenization step using a **temperature-controlled sigmoid function**, varying the **temperature hyperparameter** $\lambda$, which controls the sharpness of the relaxation.
>
> Below, we report the MAP values on the **PTC-FR** dataset under different $k/C$ ratios:
>
> |$k/C$|CoRGII (current)|CoRGII ($\lambda = 0.5$)|CoRGII ($\lambda = 1$)|
> |-|-|-|-|
> |0.1|0.17|0.18|0.17|
> |0.2|0.27|0.28|0.26|
> |0.5|0.42|0.42|0.42|
>
> We observe that the results are comparable.
>
>
> > *clarification*
>
> Line 22: Yes , it should be $G_q$; L105: $\le$ means entriwise inequality ($A \le B$) means $A[i,j] \le B[i,j]$ for all $i,j$; L129: PostingList($\tau$) is correct.

---

> > ### Comment · Reviewer_R7bo · 2025-08-04
> >
> > Thank you for the additional details. Your response has succesfully addressed all my points. In my view, the additional results on the robustness with respect to the data used to pretrain Isonet, and the theoretical and empirical runtime comparison with FHN help to understand the potential impact of CORGII.
> > One issue that seems to remain, and is also acknowledged by other reviewers, is clarity of exposition. I hope you can take this feedback into account to make CORGII more accessible, which will only help it make a broader impact. Nonetheless, I have decided to increase my score (please note that the updated score will only be visible later).

---

> ### Comment · Area_Chair_ivEK · 2025-08-04
> **Request to Engage in Reviewer-Author Discussion**
>
> Dear Reviewer R7bo,
>
> I kindly ask that you take a moment to review the author responses, including their replies to other reviewers’ comments, and acknowledge them by clicking the “Mandatory Acknowledgement” button. It is very important that you read the reviews and discuss any disagreements or concerns during this author-reviewer discussion phase so we can work toward a well-informed consensus.
>
> Thank you again for your time and contributions.
>
> Best,
> AC, NeurIPS 2025

---

### Official Review · Reviewer_d1NF · 2025-07-11

**Clarity:** 3
**Significance:** 3
**Originality:** 3
**Rating:** 5
**Confidence:** 3

**Summary:**

This paper proposed CORGII (COntextual Representation of Graphs for Inverted Indexing), which aims to improve upon currently existing graph retrieval algorithms in terms of accuracy-efficiency trade-offs. The authors borrow ideas from the text retrieval literature, proposing counterparts in the graph domain such as differentiable graph tokenization, the usage of (trainable) impact scores for query-key matching, and the usage of inverted indices. Their experimental results suggest that CorGII is able to achieve significant improvement in retrieval accuracy-efficiency trade-offs over existing methods.

**Questions:**

Please refer to the Weaknesses in the Strengths and Weaknesses section.

**Ethical Concerns:**

["NO or VERY MINOR ethics concerns only"]

**Final Justification:**

I thank the authors for preparing a detailed response to my post-rebuttal comments. They concretized the usage of "dense" vs "sparse" as essentially based on "continuous" vs "discrete" representations, and clarified what they mean by "vastly". Commendably, during this short discussion timeframe, they designed and evaluated a candidate estimator that gauges the impact of different distance metrics and justifies their usage of Chamfer distance. They also provide a discussion hypothesizing why their method, despite the high default sample complexity of contrastive learning, could potentially satisfy certain conditions that could relax the worst-case upper-bound criterion of a large number of samples, based on theoretical results from existing works. This is sufficient to address my concerns and I am thus happy to increase my score.

**Limitations:**

Please refer to the Weaknesses in the Strengths and Weaknesses section.

**Paper Formatting Concerns:**

N.A.

**Quality:**

3

**Strengths And Weaknesses:**

## Strengths:

1. The authors propose to use Chamfer distance for token matching across the discretized graphs. This brings about improved computational efficiency, and the authors argue with theoretical rationale that support the usage of non-injective mappings via the Chamfer distance in this context.

2. The authors propose an effective remedy to the degradation in recall caused by using only a single probe per query token into the indices, which is a result of using discrete node representations as opposed to continuous ones, and also, as claimed by the authors, of using Chamfer score for node alignment, via a co-occurrence expansion based mechanism from classical text retrieval.

3. The paper is very well written, with a well-justified motivation, well-organized development and description of the algorithms, as well as a systematic presentation of empirical results.

4. The authors were thorough in their empirical analysis that validate the decisions taken at most of the steps involved in the algorithm design process. Additionally, the results suggest strong improvements provided by the proposed CoRGII in terms of retrieval accuracy-efficiency trade-offs.



## Weaknesses:

1. It is not clear what is meant by "dense graph representation". How do the authors measure density in the context of graph representations?

2. A discrete, tokenized, multi-vector representation for node-wise matching, although may be able to capture more fine-grained information, seems significantly more computationally expensive than single-vector-based continuous graph representations that can return similarities with a single dot-product computation. How does the proposed CoRGII compensate for this gap? In this regard, no empirical results are provided either that compares CoRGII with single-vector-based continuous approaches in terms of computation overhead, nor is any theoretical time and space complexity analysis being provided.

3. The reason behind the use of Chamfer score instead of permutation-based node alignment causing a loss in recall is not exactly obvious. It is unintuitive as Chamfer score is also meant to capture the same alignment property.

4. Regarding the query-aware learning of the impact score using the MLP $\psi$, it is important to understand the effect of label imbalance. For node-types that occur very infrequently in the corpus, there may not be sufficiently many samples to guarantee good generalization of the learned impact score function.

Minors:

Line 387: capturi -> capture

---

> ### Author Rebuttal · Authors · 2025-07-31
>
> We would like to thank the reviewer for their comment, which will improve our paper
>
>
> > It is not clear what is meant by "dense graph representation".
>
> In text retrieval, a "sparse" index is keyed on discrete tokens. Sparse text indexes use inverted posting lists, in which the number of document (IDs) is vastly smaller than the corpus size, hence the name "sparse".
>
> In a "dense" text index, a text encoder converts text into continuous vectors and then they are indexed. Such dense representations are used for IVF, LSH (e.g., FHN) or DiskANN. Dense indexes are larger and slower to navigate than inverted indexes.
>
> > A multi-vector representation .. seems significantly more computationally expensive than single-vector-based continuous graph representations. How does the proposed CoRGII compensate ...?
>
> There are two key technical challenges in any graph search (and any information retrieval) system:
> 1. The number of corpus graphs is large, leading to impractical $\Omega(|C|)$ execution time for each query, necessitating some mechanism to **eliminate a large fraction of the corpus** at the very outset, via suitable indexing.
> 1. Suitable **scoring function design**, which will approximate the similarity between query and corpus graphs. In both text and graph retrieval, the score is frequently learnt from relevant and irrelevant items given a query.
>
> Many contemporary search systems are built as a cascade of a *retriever* (the first stage) and a *reranker* (the second stage).
>
> CORGII is fundamentally a *retriever*, which, given a query, efficiently cuts down the corpus to a much smaller *subset* of eligible candidates. How this subset is scored and reranked is orthogonal to the retriever. The computational trade-offs between single-vector and multi-vector representations that the reviewer refers to manifest primarily in the reranking stage—not in the retrieval stage where CoRGII operates.
>
> CORGII need not be hardwired to a fixed reranking policy. CoRGII is agnostic to the downstream reranker and can be seamlessly paired with any representation based on practical efficiency-accuracy requirements.  Currently CORGII is cascaded with a multi-vector-based ranker (IsoNet). Instead, it can be paired with a single-vector-based reranker.
>
> Indeed, multivector based score computation is slower than single vector based score computation, but CorGII builds an efficient retrieval system that supports faster retrieval than even single vector based dense retrieval.
>
> Specifically, instead of indexing single dense vector representations and probing the index with multiple dense query vectors, CORGII uses a subsequent encoder step to discretize dense vectors to sparse representations, making graphs appear like documents (with synthetic tokens), so that efficient inverted indices can be used. As with text inverted indices,  queries are discrete tokens and not vectors.
>
> > *In this regard, no empirical results, theoretical  complexity analysis being provided.*
>
> We fix MAP at target levels $m^*$ and compare retrieval **time** required by different pipelines on **PTC-FR**, while maintaining comparable accuracy.
>
> First, we compare CoRGII against FHN, showing that CoRGII achieves the same MAP up to 2× faster than FHN:
> |$m^*$|FHN| CoRGII|
> |-|-|-|
> |0.38|0.06s|0.04s|
> |0.40|0.10s|0.05s|
> |0.45|0.12s|0.08s|
>
> Next, we compare CoRGII with multi-vector reranking against CoRGII with single-vector reranking. While single-vector reranking is faster per comparison, it requires retrieving a much larger portion of the corpus to achieve even moderate MAP values (e.g., 0.38), resulting in higher overall latency. Moreover, due to its limited expressiveness, single-vector reranking fails to reach higher accuracy regimes altogether.
> |$m^*$|CoRGII + Multi-vector reranking| CoRGII + Single-vector reranking
> |-|-|-|
> |0.38|0.06s|0.08s|
> |0.40|0.10s|Never reaches this MAP|
> |0.45|0.12s|Never reaches this MAP|
>
> Thus, CoRGII + Multi-vector reranking achieves the best trade-off. While multi-vector scoring is slower per candidate, the number of candidates it needs to consider is substantially lower due to its higher precision. This leads to fewer pairwise computations overall—making it faster in practice than using single-vector representations to achieve the same accuracy.
>
> Notably, CORGII is such an efficient and high-quality retriever compared to the baselines that it is able to compensate the slow reranking time of the multivector methods, resulting in the faster retrieval+ranking time than the baselines including those using faster single-vector-reranker.
>
> We understand that multivector based reranking is slower than single vector based reranking. However, in contrast, multivector guided retriever, especially inverted indices  like CORGII is much faster than single vector guided retriever, like FHN. We provide a theoretical analysis of this as follows:
>
> Suppose the vocabulary size is $M$. Posting list lengths typically follow Zipf's law, as we see in the literature and also shown in our problem in App F.4 (page 25, Appendix). Hence, by definition of Zipf law, the probability distribution of occurrence of the $k$th most frequent token (in the corpus) is $f(k) = \frac{A}{k^{\delta}}$ with $0.5<\delta < 1$.
> $$\int _{1} ^M f(k)dk = 1 \implies A\frac{M^{1-\delta}}{(1-\delta)} - \frac{A}{1-\delta}=1$$
> Since $M$ is large, we have $$A = O(M^{\delta-1})$$
> Let the aggregated length of posting lists is $L$. Then, assuming the $M$-th popular token will occur only once, we have:
> $$ \frac{LA}{k^{\delta}} = 1 \text{ at } k = M$$
> This implies $L = O(M)$. For one query token, the expected number of corpus graphs returned will be expected length of posting list under Zipf law. For a whole query, the expected number of retrieved graphs is $|V_q| \mathbb{E} (\text{Length of posting list})$, where
> $$\mathbb{E} (\text{Length of posting list}) = \int _{1} ^{M} Lf(k) \times f(k) dk = O(M M^{2\delta-2})  \int _{1} ^{M} {k^{2\delta}} dk$$
> Now, $\int _{1} ^{M} {k^{2\delta}} dk =$ constant since $\delta\in (0.5,1)$. Therefore, expected posting list length is $O(M^{2\delta-1}) \in [O(\sqrt{M}),O(M)].$ Hence, the expected time complexity of CORGII
> is $O(M^{2\delta-1})$. Similarly, the space complexity of index is the total length of posting list $O(L) = O(M)$
>
> In contrast, for the single vector representation in FHN, the seminal paper of (Indyk, Motwani, 1998) showed that the time complexity is $O(|C|^{\gamma})$ where $\gamma<1$ is the efficiency parameter of LSH. The space complexity is $O(|C|^{1+\gamma})+O(|C|)$.
>
> Since in practice, number of corpus items is much more larger than size of the vocabulary, i.e., $|C|\gg M$, CORGII is more efficient than single vector retrieval in terms of time and accuracy. Moreover, this is consistent with text based retrieval where inverted index is widely adopted over single vector based indexing.
>
> We also report index construction time and memory usage, both of which remain modest and practical across datasets, supporting the scalability of CORGII.
> |Indexing time|CoRGII|
> |-|-|
> |PTC-FR|29s|
> |COX2|27.2s|
>
> |Indexing memory|CoRGII|
> |-|-|
> |PTC-FR|3.6M|
> |COX2|4.3M|
> >*The reason behind the use of Chamfer score instead of permutation-based node alignment ... not exactly obvious.*
>
> When  $G_q\subseteq G_c$, we would want that $Z _q := (z _q(u) : u\in V _q )= Z _c :=(z _c(v) : v\in V _c)$, to ensure exact token matching in inverted index for the relevant query corpus pairs. To compute the matching distance between the sets of tokens, it would have been natural to adopt permutation which gives us the normed distance $||Z _q - P Z _c || _1$. Now, if $P$ is a hard permutation matrix and provides an accurate alignment between query and corpus, then indeed the above normed distance could have been used to train.
>
> But, in practice, any graph retrieval model is trained under distance supervision of relevance labels $y _{qc}$ and therefore, the ground truth alignments are not known a-priori. Consequently, $P$ is typically modeled as a doubly stochastic soft permutation matrix and trained end-to-end. While this approximation enables end-to-end differentiability, the continuous nature of $P$ smears the values in $Z _q$ and $Z _c$, leading to poor discretization.
>
> To ensure a high quality discretization, we avoid soft permutation based approximation and seek a **discrete approximation** of the permutation. In this case, $P _{u,u'} \approx \mathbf{1} [ argmin _{v} ||z _q (u) - z _c (v)|| _1==u']$, where $\mathbf{1}( . )$ is an indicator function. This readily gives $||Z _q - P Z _c || _1$ to be the same as Chamfer score. Instead of continuous approximation, Chamfer score provides a discrete approximation, which is amenable for training z since it is a discrete embedding.
> >... For node-types that occur very infrequently in the corpus, there may not be sufficiently many samples to guarantee good generalization of the learned impact score function.
>
> Even if there are rare graphs, we are able to train impact score, because we are using ranking loss on the aggregated ranked list. Suppose, we have  relevant query corpus pairs G_q and G_c each with rare tokens. In batching, our ranking loss ensures that $S(G _q, G _c) > S(G _{q'}, G _{c'})$ whenever $G _{q'}$ and $G _{c'}$ are not relevant, including graphs which have rich tokens. This allows the trainer to feed sufficient signal into the impact scorer. If we use $S(G _q,G _c)$ as the sole scoring function, then we obtain a modest MAP~0.28 even if $S$ is not any graph matching model and is driven by a simple MLP.
>
> This can be further enhanced by using the following modification of the impact scores
> $$\sum  _{u \in V _q} Impact _{\psi}(\hat{z}  _q(u), h  _q(u))
>  \mathbf{1}(\hat{z}  _q(u) \in \omega(G _c)) \to \sum _{u \in V _q}  \lambda  Impact _{\psi}(\hat{z}  _q(u), h _q(u))
>  \mathbf{1}(\hat{z}  _q(u) \in \omega(G _c))$$
> Here, $\lambda$ is a hyperparameter which is more than 1 and is only applied if $\hat{z}  _q (u)$ is a rare token.

---

> ### Comment · Reviewer_d1NF · 2025-08-05
> **Response to Rebuttal**
>
> I thank the authors for taking my comments into account and preparing a rebuttal, which addresses a number of my concerns. Below, I provide my response to their rebuttal.
>
> 1. So, is the difference between dense vs sparse in the usage of discrete vs continuous-valued tokens? In their response, the authors mention that in sparse indices, the number of document (IDs) is "vastly" smaller than the corpus size. "Vastly" is a relative term and is not quantitatively defined. How big of a difference does there need to be in order to qualify as vast? For instance, in LSH, the number of hash buckets can be chosen to also be vastly smaller than the corpus size, but this does not make LSH a sparse indexing method. Although I get the sense, it is important to be precise about these terminologies. Further, I understand the usage of sparse vs dense indices in the context of text. However, the authors borrow this idea from text retrieval and apply it to graphs, so it needs to be defined in the context of graphs too.
>
> 2. The provided explanations, supported by the empirical and theoretical results showing the accuracy-efficiency trade-off of CORGII in conjunction with single and multi-vector reranking addresses my concerns.
>
> 3. The motivation behind the usage of Chamfer distance is now clear. However, it would be good to have an estimate on the error that is incurred due to conversion from continuous to discrete, bit-like representations, or at least a proposed strategy on how an estimator might be designed for this purpose. Otherwise, it is difficult to gauge a priori the downstream impact of this conversion in the form of lost recall.
>
> 4. I am not entirely convinced by the strategy that the authors proposed for learning the impact score function with rare label types. Although the authors propose a canonical way of upweighting the impact function for rare label types, a common practice for dealing with label imbalance, it is well known that simple reweighting often does not resolve the issues posed by imbalance [1, 2, 3]. Additionally, contrastive learning (which is essentially the form that the proposed ranking loss admits) is known to have high sample complexity with increasing dimensionality [4]. I am not looking for the authors to provide solutions to these in this work as they could constitute independent research questions, but there needs to be discussions on these as potential challenges / limitations.
>
> In view of these, I presently choose to retain my score.
>
> References:
>
> [1] Duggal et al., "HAR: Hardness Aware Reweighting for Imbalanced Datasets", IEEE International Conference on Big Data 2021.
>
> [2] Yu et al., "A Re-Balancing Strategy for Class-Imbalanced Classification Based on Instance Difficulty", CVPR 2022.
>
> [3] Luo et al., "Revive Re-weighting in Imbalanced Learning by Density Ratio Estimation", NeurIPS 2024.
>
> [4] Alon et al., "Optimal Sample Complexity of Contrastive Learning", ICLR 2024.

---

> ### Author Response · Authors · 2025-08-07
> **Response to reviewer d1NF (1/2)**
>
> Dear reviewer d1NF,
>
> We thank you for your interest and detailed response to our rebuttal. We agree that the issues you raised require further clarification, which we provide below.
>
> > (1) Dense vs Sparse; "vastly"
>
> We agree with you that “vastly” was imprecise and should be avoided. Note that an index built on a corpus is described as “sparse” or “dense” depending on the way in which the corpus is presented to the indexer and not whether the index itself is sparse or dense.
>
> Suppose the raw textual vocabulary size is $10^4$ tokens, and there are $10^7$ documents. Then the document-term matrix is $10^7 \times 10^4$, with $10^{11}$ elements. Each document may typically have very small number of terms (by Zipf law, as also discussed in our theoretical analysis). This results in “sparse” matrix. However, we could not find any folklore threshold of sparsity in the Information Retrieval literature. An inverted index built from this matrix is traditionally called a “sparse index” (but sparsity arises from the source: the sparse input matrix).
>
> Now suppose each document is converted into a 768-dimensional real vector using BERT. All elements in such a vector are expected to be nonzeros. The corpus can now be represented using a $10^7 \times 768$ matrix, which is completely filled with nonzeros, and therefore classically called a “dense” matrix. The columns of this matrix are completely dense. An approximate nearest neighbor (ANN) index built from this matrix (be it an LSH, or a HNSW) is called a “dense index”. One cannot gainfully build an inverted index from this matrix using columns as pseudo-"words" because every pseudo-word would have occurred in every document.
>
> In the context of graphs, our usage of the term “sparse indexing”,  inherits from the conventional usage in text retrieval,  where it refers to discrete token-based lookup structures such as inverted indices. In contrast, “dense indexing” operates over continuous vector spaces, such as might be obtained as direct outputs of a GNN.
>
> > (3) However, it would be good to have an estimate on the error... gauge a priori the downstream impact
>
> Final recall computation requires impact score and multiprobing. We agree that, it would be useful to gauge the effect of Chamfer before going to that stage. Toward that goal, we provide the following error based estimator to evaluate the efficacy of Chamfer without measuring recall.
>
> We compute ground-truth distance $\text{dist}^*$, distance $\text{dist} _{\text{soft}}$ under soft permtuation and Chamfer distance $\text{dist} _{\text{chamfer}}$ (as defined in the paper, see equation 4 there):
>
> $\text{dist}^* =  \min _{\mathbf{P}} \sum _{i, j} \big[ \mathbf{A} _q - \mathbf{P}  \mathbf{A} _c\mathbf{P}^{ T} \big] _+[i, j]$
>
> $\text{dist} _{\text{soft}} = \sum _{i, j} \big[ \mathbf{Z}  _q - \mathbf{P}^{\text{soft}}\mathbf{Z}  _c \big] _+ [i, j]$
>
>
> The following table shows that $\text{dist}^*$ is better approximated by $\text{dist} _{\text{chamfer}}$ than by $\text{dist} _{\text{soft}}$.
>
> | Dataset | Average($\text{abs}(\text{dist}^* - \text{dist} _{\text{soft}})$) | Average($\text{abs}(\text{dist}^* - \text{dist} _{\text{chamfer}}))$ |
> | -------- | -------- | -------- |
> | PTC-FR   |   26.9 |   **20.18**  |
> | PTC-FM   |  35.42 |  **22.53**   |

---

> ### Author Response · Authors · 2025-08-07
> **Response to reviewer d1NF (2/2)**
>
> > (4) Discussion on label imbalance, sample  complexity
>
> Many thanks for pointing out these references [A-D], listed below. We can adapt some of these proposals to enhance impact score computation as follows. Consider the resampling or re-balancing algorithm proposed in [A]. At each iteration $t$, we use a sampling procedure, formulated as a multinomial distribution Multinomial$(D ^t)$ over all corpus items (with $D^t = [D _c ^t : c\in C]$) over difficulty estimates $D _c ^t$ of tokenizing the graphs arising due to presence of rare tokens. Then, we re-train or fine-tune the tokenizer GTNet on graphs sampled from this distribution. In this case, we can compute predictions of tokens as $z^{(t)} (u)$ for each node $u$. Then, we leverage the prediction variations $z^{(t)}(u)-z^{(t-1)}(u)$ (and their signs) to compute difficulty for each node. Then, we can aggregate these difficulty across nodes to compute difficulty $D _c ^{(t)}$ for $G _c$. We can also compute $\text{Impact}(z ^{(t)} (u), h)\cdot\mathbf{1}(z ^{(t)} (u) \in \omega(G _c))-
> \text{Impact}(z ^{(t-1)} (u), h) \cdot \mathbf{1}(z ^{(t-1)} (u) \in \omega(G _c))$ as a proxy difficulty per node. This would be a more principled alternative to our earlier proposal involving $\lambda$.
>
> The paper [B] assigns suitable weights $\alpha(x,y)$ on the loss per each instance. It would be an interesting research problem to design $\alpha(z (u) , G _c)$, similar to [B]. A key challenge in immediately adapting [B] is that we may need to estimate $P(G _c | z(u))$ as used in Eq. (2) in [B], which would pose an interesting research question.
>
>
> **Discussion of Alon et al. [C]** Contrastive training is ubiquitous in dense retrieval, starting from such classic papers as DPR [Karpukhin et al., 2020 arXiv:2004.04906], ANCE [Xiong et al., 2021 arXiv:2007.00808], and RocketQA [Qu et al., 2021 arXiv:2010.08191]. Further enhancements were reported in COIL [Gao et al., 2021 arXiv:2104.07186], SimCSE [Gao et al., 2021 arXiv:2104.08821] and GTR [Ni et al., 2021 arXiv:2112.07899]. Alon et al. [C] provide extremely insightful observations on the sampling complexity of contrastive learning from a PAC point of view.   They show that, for contrastive training, with $n$ points in $d$ dimensionl space, $\widetilde{\Omega}(\min\{nd, n^2\})$ samples are needed. They do provide two escape clauses, however. First, the number of samples needed reduces if the points admit some underlying clusters. Second, if each triplet comes with a demand for large separation, then the lower bound drops to $\widetilde{\Omega}(n)$. In LSH-style retrieval, large separation is usually the goal, in the sense that points “near” the query should fall into the same bucket (Hamming distance 0) as the query, and “far” points (that are $\ge (1+\epsilon)$ times farther than near points) should fall into other buckets (Hamming distance $\ge 1$). It is possible that this is what makes contrastive training work well in retrieval applications. It would be of great interest to collect empirical evidence for or against this hypothesis.
>
> We thank the reviewer for pointing us to these intriguing recent works [A,B,C,D], and we will be sure to include a discussion in our manuscript (Neurips allows an extra page in camera ready. Therefore, we plan to include this discussion in the main, if our paper gets accepted)
>
> [A] Yu et al., "A Re-Balancing Strategy for Class-Imbalanced Classification Based on Instance Difficulty", CVPR 2022.
>
> [B] Luo et al., "Revive Re-weighting in Imbalanced Learning by Density Ratio Estimation", NeurIPS 2024.
>
> [C] Alon et al., "Optimal Sample Complexity of Contrastive Learning", ICLR 2024.
>
> [D] Duggal et al., "HAR: Hardness Aware Reweighting for Imbalanced Datasets", IEEE International Conference on Big Data 2021.

---

> > ### Comment · Reviewer_d1NF · 2025-08-08
> > **Response to Authors #2**
> >
> > I thank the authors for preparing a detailed response to my post-rebuttal comments. They concretized the usage of "dense" vs "sparse" as essentially based on "continuous" vs "discrete" representations, and clarified what they mean by "vastly". Commendably, during this short discussion timeframe, they designed and evaluated a candidate estimator that gauges the impact of different distance metrics and justifies their usage of Chamfer distance. They also provide a discussion hypothesizing why their method, despite the high default sample complexity of contrastive learning, could potentially satisfy certain conditions that could relax the worst-case upper-bound criterion of a large number of samples, based on theoretical results from existing works. This is sufficient to address my concerns and I am thus happy to increase my score.

---

### Official Review · Reviewer_WWoV · 2025-07-20

**Clarity:** 3
**Significance:** 3
**Originality:** 4
**Rating:** 5
**Confidence:** 2

**Summary:**

The paper presents CORGII, a novel framework that integrates dense neural graph embeddings with scalable, classical retrieval mechanisms. CORGII proposes contextual tokenization of graphs through a differentiable process that converts node-level representations into discrete binary tokens mapped to a latent vocabulary.
These tokens are used to construct efficient inverted indices similar to those in text retrieval, enabling
scalable subgraph search and retrieval with fast candidate generation. Unlike prior dense approaches that
require exhaustive scoring, CORGII supports a trainable impact scoring scheme and a multi-probing strategy that balances accuracy and efficiency. The approach shows strong empirical results on multiple datasets and outperforms recent baselines in both precision and latency

**Questions:**

- Could the authors provide a qualitative or visual analysis of the token distribution over the graph space? Are tokens reused across semantically similar subgraphs?
- What is the trade-off between token vocabulary size and retrieval efficiency? Is there a trade-off point?
- How would CORGII handle temporal graphs, where node connectivity evolves?
- Can the impact scoring scheme be regularized or interpreted in ways that reveal important graph patterns or regions?
- Are there graph types or domains where contextual tokenization fails to outperform dense or exact
methods?
- Is the framework compatible with privacy-preserving graph representation ?

**Ethical Concerns:**

["NO or VERY MINOR ethics concerns only"]

**Final Justification:**

The responses from the authors completely cleared all my concerns. I updated my initial score from borderline accept to accept since the paper is solid and valid.

**Limitations:**

yes

**Paper Formatting Concerns:**

no one

**Quality:**

3

**Strengths And Weaknesses:**

Strengths:

- CORGII successfully bridges classical IR techniques with graph neural networks, enabling scalable and
interpretable graph retrieval.
- The core idea of contextual tokenization, inspired by language tokenization, is original and well-justified for representing graph structures sparsely.
- The use of a differentiable tokenizer enables an end-to-end training process, ensuring that tokenization is aligned with retrieval goals rather than static feature engineering.
- Incorporating impact scores adds flexibility to the ranking process, thus making it more adaptive than heuristic scoring schemes.
- The experimental results are strong, demonstrating consistent  and visible improvements in retrieval precision and
latency.

Weaknesses:
- Multi-probing is used effectively to balance trade-offs, but more theoretical or empirical exploration of
probing strategies would strengthen the work.
- Some technical sections are dense, and concepts like latent vocabulary structure, token reuse, or token
collisions could be elaborated for better accessibility.
- There is a reliance on pre-trained GNNs and clarifying the extent to which performance depends on pretraining quality would be beneficial.
- Additional benchmarks, especially on very large or noisy graphs, could help validate scalability claims.
- Lack of interpretability in the learned token representations and impact scores limits their usability in critical decision-making settings.

---

> ### Author Rebuttal · Authors · 2025-07-31
>
> We thank the reviewer for the comments, which will improve our paper.
>
> > *empirical exploration of probing strategies.*
>
> We experiment with various multiprobing strategies. Specifically, we compare the cooccurrence multiprobing (CM) strategy with a traditional Hamming distance-based multiprobing variant (HM). In HM, we probe the inverted index using not only $\tau$, but also nearby tokens within a Hamming ball of radius $r$ in the binary space $(0,1)^D$. Thus, HM enables token matching by leveraging proximity in Hamming space. We also experiment with a single-probing strategy, which is equivalent to setting $r=0$. The table below shows MAP values for different $r$ and $b$ for the PTC-FR dataset. $b$ is the number of similar tokens in the CM strategy. Higher $r$ and $b$ indicate a more diffused neighborhood and more candidates.
>
> |$k/C$|Single-Probe(r=0)|HM,r=1|HM,r=3|HM,r=7|CM,b=8|CM,b=32 |CM,b=64|
> |---|------------------|------|------|------|------|--------------|--------|
> |0.1|0.16|0.16|0.16|0.12|0.15|0.17|0.18|
> |0.2|0.25|0.25|0.25|0.21|0.24|0.27|0.28|
> |0.3|0.31|0.31|0.32|0.29|0.30|0.34|0.34|
> |0.4|0.36|0.36|0.36|0.25|0.35|0.38|0.39|
> |0.5|0.38|0.39|0.40|0.39|0.39|0.42|0.42|
> |0.6|0.48|0.41|0.42|0.43|0.42|0.44|0.44|
> |0.7|0.48|0.48|0.44|0.45|0.44|0.46|0.46|
> |0.8|0.48|0.48|0.48|0.46|0.46|0.47|0.47|
> |0.9|0.48|0.48|0.48|0.48|0.47|0.48|0.48|
> |1.0|0.48|0.48|0.48|0.48|0.48|0.48|0.48|
>
> We see that single probing cannot trade off between accuracy-efficiency effectively.  CM consistently achieves better trade-off
> than the corresponding variant of HM and single-probing strategy, while smoothly spanning the full range of retrieval selectivity.
>
> > *Some technical sections are dense, and concepts like latent vocabulary structure, token reuse, or token collisions could be elaborated for better accessibility.*
>
> In sparse indices, the indexed tokens are interpretable words. In CORGII, tokens are not interpretable, but the expectation is that distinctive subgraph motifs will be mapped to the same or strongly correlated tokens. We cannot find usage of the phrases "token reuse" or "token collisions" in our manuscript.
>
> > *pre-trained GNNs ... clarifying the extent to which performance depends on pretraining quality would be beneficial.*
>
> We clarify that our **GNN is not pre-trained**; it is trained independently for indexing, as described in lines 165–167 and illustrated in Figure 1. Consequently, the tokenization network (GTNet) does not rely on any pre-trained embeddings.
> Since the tokenizer operates without any pre-trained GNN, the resulting index ― built solely on the tokens ― is also independent of pre-trained embeddings.
>
> Only our impact scoring model does utilize pre-trained embeddings from IsoNet. However, this results in minor dependence of pre-trained embeddings. We illustrate this by using two alternatives of current pre-trained embeddings from IsoNet:
> 1. Instead of pre-trained embeddings $h$,  we use $x$ to as input to the impact scoring model (Impact($\tau,x$), Eq. 8).
> 1. Instead of training $h$ on the same dataset, we train $h$ some other dataset and then transfer cross-domain embeddings into the impact scoring model. Specifically, we train h on the PTC-FM dataset and apply it to PTC-FR, and likewise train on PTC-MR and apply to COX2.
>
> Following tables show the efficiency in terms of fraction of graphs $k/C$ retrieved to achieve atleast a certain MAP=$m^*$ for PTC-FR (first table) and COX2 (second table) datasets. Lower k/C is better and indicates higher efficiency.
>
> |$m^*$|Our (current) | Our (use $x$)|Our ($h$ trained on other dataset)|FHN|DiskANN-multi|IVF-multi|
> |-|-|-|-|-|-|-|
> |0.38|0.38|0.376|0.42|0.51|0.875|0.762|
> |0.40|0.44|0.436|0.47|0.81|0.875|0.896|
> |0.45|0.645|0.626|0.69|0.99|0.935|0.964|
>
> |$m^*$|Our (current) | Our (use $x$)|Our ($h$ trained on other dataset)|FHN|DiskANN-multi|IVF-multi|
> |-|-|-|-|-|-|-|
> |0.38|0.31|0.425|0.27|0.61|0.552|0.644|
> |0.40|0.35|0.482|0.29|0.71|0.821|0.644|
> |0.45|0.45|0.526|0.37|0.99|0.821|0.833|
>
> The efficiency does not change much and even if we do not use pre-trained embeddings, our method outperforms baselines.
>
> > *... validate scalability claims.*
>
> We perform experiments with noisy graphs, which results in $\sim 10\\%$ noise in the embedding space.
> We report the MAP values achieved at different retrieval fractions $k/C$ for various hashing-based methods for PTC-FR (first table) and COX2 (second table) datasets. Higher MAP is better. We observe that our method consistently achieves stronger retrieval quality across a range of candidate pool sizes, demonstrating superior efficiency and robustness compared to the baselines.
>
> |  $k/C$ | CoRGII  | FHN | DiskANN-multi | IVF-multi |
> |-----|---|----|------|------|
> |0.1|0.18|0.15|0.10|0.10|
> |0.2|0.28|0.25|0.16|0.16|
> |0.3|0.35|0.34|0.16|0.16|
> |0.4|0.38|0.34|0.21|0.23|
> |0.5|0.42|0.38|0.37|0.30|
> |0.6|0.44|0.38|0.37|0.30|
> |0.7|0.46|0.42|0.37|0.48|
> |0.8|0.47|0.42|0.44|0.48|
> |0.9|0.48|0.48|0.46|0.48|
> |1.0|0.48|0.48|0.48|0.48|
>
> |$k/C$| CoRGII | FHN  | DiskANN-multi  | IVF-multi |
> |---|-----|-----|------------|-----------|
> |0.1|0.22|0.14|0.19|0.10|
> |0.2|0.32|0.23|0.19|0.19|
> |0.3|0.39|0.33|0.29|0.27|
> |0.4|0.44|0.33|0.35|0.32|
> |0.5|0.48|0.34|0.39|0.36|
> |0.6|0.51|0.46|0.51|0.40|
> |0.7|0.53|0.46|0.51|0.56|
> |0.8|0.55|0.48|0.51|0.56|
> |0.9|0.56|0.53|0.55|0.56|
> |1.0|0.56|0.56|0.56|0.56|
>
> We also perform experiments with large number of graphs (10^6). It takes 272s to index and 5.44 GB GPU memory (or 19.4 GB CPU memory). Hence, it can be easily run in a very lightweight server.
>
> > *Regularization, Interpretability*
>
> To examine interpretability of learned token, we compute $(||\hat{z} _u- \hat{z} _v|| _1) _{u,v}$ ― a list of pairwise Hamming distance of discrete token embeddings $\hat{z}$ between nodes. Similarly we obtain a list of distance between the $L$-hop neighborhoods of the nodes in the Eucledian space. We observe the rank correlation between these lists is 0.71, which indicate the token embeddings capture the structures.
>
>
> We can impose a regularizer to enhance more sparsity in $\hat{z}$ embeddings, which can control the above rank correlation for PTC-FR.
>
> |L1 norm of \hat{z}|rankCorr |
> |-|-|
> |2.99|0.71|
> |3.27|0.62|
> |3.50|0.61|
>
> Our impact score is also trained to assign higher score to relevant graph $G _{c}$ to a given query $G _q$, even if $G _c$ is not relevant to other queries. We explicitly use ranking loss rather than classification loss to ensure this, which encourages $S(G _q, G _{c+})> S(G _q, G _{c-}) +m$. We tried with m=0.1, 0.01 and 1 and observed the best performance m =0.1. If we use $S(G_q,G_c)$ as the sole scoring function, then we obtain a modest MAP~0.28 even if $S$ is not any graph matching model and is driven by a simple MLP.
>
> > *qualitative or visual analysis of the token distribution?*
>
> In Fig. 14 in Appendix, we showed this qualitative analysis.
> Here, we show a small corresponding table, which shows |Token(G)|, the size of the unique token set of corpus graph G vs $\rho(G)$, the rank of graph G when sorted in
> descending order of unique token count for PTC-FR.
>
> |Token(G) size |$\rho(G)$|
> |-|-|
> |14|5K|
> |13|10K|
> |12|20K|
> |11|30K|
> |10|40K|
> |9|60K|
> |8|80K|
> |6|90K|
> |2|100K|
>
> Similar to text, it shows that that most documents have a small number of unique tokens. More details are in App F.4 (page 25, Appendix)
>
> > *Are tokens reused across similar subgraphs*
>
> As mentioned before, we compare the list of pairwise hamming distance between discrete tokens $(||\hat{z} _u- \hat{z} _v|| _1) _{u,v}$ and the distance between the $L$-hop neighborhoods of the nodes in the continuous embedding space. We observe the rank correlation between these lists is 0.71, which indicate the token capture similar subgraphs.
>
>
> > *What is trade off between vocabulary size and retrieval efficiency*
>
> We explore this trade-off by varying the vocabulary size, which directly affects the average posting list length-a key factor governing retrieval efficiency. The table below reports MAP values for three configurations in PTC-FR.
>
> |Average Posting List Length|MAP|
> |-|-|
> |1642.9|0.31|
> |1792.28|0.31|
> |965.4|0.29|
>
> There is a trade off between accuracy and efficiency (inverse to fraction of retrieved graphs). Notably, the configuration with a posting list length of ~965 retrieves approximately half as many candidates as the larger configurations, while incurring only a minor drop in MAP, suggesting this setting strikes a favorable balance between speed and performance.
>
>
> >*How would CORGII handle temporal graphs, where node connectivity evolves?*
>
> This is an excellent branch-out for future work. Much depends on the rate of change. If the graph can change while the query is executing, perhaps little can be done. For slower rates, given CORGII turns graphs into documents in discrete term space, classic literature on mutable corpora can be harnessed.
>
> [Moffat1996] [Managing Gigabytes: Compressing and Indexing Documents and Images]
>
> [Buttcher2006] [Hybrid index maintenance for growing text collections]
>
> [Lester2005] [Fast On-Line Index Construction by Geometric Partitioning]
>
> There are classic ideas available to support corpus mutation: Log-structured merge trees (LSM), segment-based indexing, buffering and batched updates, deletion markers, and index snapshots and versioning.
>
> > *Is the framework compatible with privacy-preserving  representation?*
>
> We believe that the discretization scheme allows some obfuscation to protect privacy. Currently, the discretization is deterministic. However, we can use sample the entries in discrete embeddings using a Bernoulli distribution:  instead of $\hat{z} _u =  \mathbf{1}(sigmoid(x_u) >0.5)$, we can use $\hat{z} _u  = \text{Bernoulli}(sigmoid(x_u/\lambda))$. This will allow us to ensure privacy through differential privacy more formally, where the amount of privacy will be controlled by $\lambda$.

---

> ### Author Response · Authors · 2025-08-01
> **Few clarifications**
>
> Dear reviewer WWoV,
>
> We discovered minor errors in some rebuttal tables. Due to the image restriction, we converted scatter plot pickle files into tables using a script. A small bug was found and fixed near the deadline, but we accidentally pasted the older version of the table in the rebuttal.
>
> Therefore, we are reaching out to you immediately after the discussion windows open.
>
>
> For clarity, we present the erroneous and corrected results in response to *An empirical exploration of probing strategies* below:
>
> Recall that all the numbers are the maximum MAP reached until the given k/C value (PTC-FR)
>
> |$k/C$|Single-Probe(r=0)|HM,r=1|HM,r=3|HM,r=7|CM,b=8|CM,b=32 |CM,b=64|
> |---|------------------|------|------|------|------|--------------|--------|
> |0.1|0.16|0.16|0.16|0.12|0.15|0.17|0.18|
> |0.2|0.25|0.25|0.25|0.21|0.24|0.27|0.28|
> |0.3|0.31|0.31|0.32|0.29|0.30|0.34|0.34|
> |0.4|0.36|0.36|0.36|0.25|0.35|0.38|0.39|
> |0.5|0.38|0.39|0.40|0.39|0.39|0.42|0.42|
> |0.6|~0.48~ 0.38|0.41|0.42|0.43|0.42|0.44|0.44|
> |0.7|~0.48~ 0.38|~0.48~ 0.41|0.44|0.45|0.44|0.46|0.46|
> |0.8|~0.48~ 0.38|~0.48~ 0.41|0.48|0.46|0.46|0.47|0.47|
> |0.9|~0.48~ 0.38|~0.48~ 0.41|0.48|0.48|0.47|0.48|0.48|
> |1.0|0.48|0.48|0.48|0.48|0.48|0.48|0.48|
>
>
> These corrected results further reinforce the importance of Cooccurrence Multiprobing (CM). Notably, Appendix F.1 already shows these correct numbers in figure form, where Single Probe and HM (r=1) fail to improve MAP beyond 0.38 and 0.41 after k/C = 0.6 and 0.7, respectively.
>
>
>
> In response to *... validate scalability claims.*:
>
> **PTC-FR**
> |  $k/C$ | CoRGII  | FHN | DiskANN-multi | IVF-multi |
> |-----|---------------|--------------|-------------------------|--------------------|
> |0.1|0.18|0.15|0.10|0.10|
> |0.2|0.28|0.25|0.16|0.16|
> |0.3|0.35|0.34|0.16|0.16|
> |0.4|0.38|0.34|0.21|0.23|
> |0.5|0.42|0.38|0.37|0.30|
> |0.6|0.44|0.38|0.37|0.30|
> |0.7|0.46|0.42|0.37|~0.48~ 0.38|
> |0.8|0.47|0.42|0.44|~0.48~ 0.46|
> |0.9|0.48|0.48|0.46|~0.48~ 0.47|
> |1.0|0.48|0.48|0.48|0.48|
>
> **COX2**
> |$k/C$| CoRGII | FHN  | DiskANN-multi  | IVF-multi |
> |-----|---------------|--------------|-------------------------|--------------------|
> |0.1|0.22|0.14|0.19|0.10|
> |0.2|0.32|0.23|0.19|0.19|
> |0.3|0.39|0.33|0.29|0.27|
> |0.4|0.44|0.33|0.35|0.32|
> |0.5|0.48|0.34|0.39|0.36|
> |0.6|0.51|0.46|0.51|0.40|
> |0.7|0.53|0.46|0.51|~0.56~ 0.41|
> |0.8|0.55|0.48|0.51|~0.56~ 0.49|
> |0.9|0.56|0.53|0.55|~0.56~ 0.54|
> |1.0|0.56|0.56|0.56|0.56|
>
> These results show that our method is robust to noisy graph inputs.
>
> Please also note that we had initially misunderstood what you had meant by token reuse and collisions. We now understand that you mean token collision by token matching. We fully agree with you. If our paper gets accepted, we will get one extra page, where we can elaborate the preliminaries and will also expand Appendix.

---

> > ### Comment · Reviewer_WWoV · 2025-08-04
> > **Response to authors' rebuttal**
> >
> > Dear Authors, I really appreciated the amount of work you did to address my concerns.
> >
> > I apologize for having misunderstood the pre-trained stuff. Now your response is clear and your ablation study permits to better highlight the goodness of your proposal.
> >
> > Yes, about tokens, it would be helpful to have additional materials to better explore this aspect.
> >
> > I do not have other comments, since the authors clarified almost all my comments.

---

### Author Response · Authors · 2025-08-04
**Some numbers on Scalability**

Dear reviewers,

Timing analysis (esp. for the baseline) across some datasets (including 1M corpus size) required longer time and access to server free from other processes, otherwise time calculation will be inaccurate. As a result, some numbers were not ready before the deadline.

We observed that in PTC-FM, obtaining MAP 0.36, 0.37, 0.40 requires 0.08s, 0.12s, and 0.13s for FHN and 0.03s, 0.04s, and 0.045s for CoRGII, respectively (per query). To output the same fraction of retrieved graphs for a 1M corpus size, FHN takes 1.02s, 1.63s, and 1.64s, whereas CoRGII takes only 0.5s, 0.54s, and 0.65s (per query).  These numbers suggest that CorGII is 2x–3x faster, even for 1M corpus size.

---

### Author Response · Authors · 2025-08-04
**Confused about mandatory acknowledgement notifications**

Dear AC and reviewers,
We have received mandatory acknowledgement notifications by some reviewer(s) (which includes a statement that they have engaged in discussions with authors) but we have no record of any discussion of our rebuttal. Therefore we are confused and would like to ensure we are doing our part and not missing any vital step or messages from you.

---

### Author Response · Authors · 2025-08-09
**Summary of rebuttal**

We take this opportunity to thank the reviewers for their constructive feedback and for engaging deeply with our work. Their comments have helped us clarify our exposition, enhance our empirical analysis, and further highlight the novelty and practical relevance of CoRGII.

During the rebuttal, we have addressed the reviewers’ concerns and strengthened the paper in the following ways:

- Clarified the distinction between $\mathbf{x}(u)$ and $\mathbf{h}(u)$, detailing their respective roles in CoRGII and ensuring the architecture description avoids ambiguity and discussed the effect of backbone model.

- Clarified efficiency of our method against FHN in terms of  runtime comparison, indexing time and memory construction costs.

- Discussed a theoretical complexity analysis of CoRGII’s retrieval pipeline, showing that the token-based inverted index enables sub-linear query time in the corpus size under realistic posting list distributions, and that multi-probing adds only a constant-factor overhead relative to single-probe retrieval.

- Summarized and clarified key ablation studies for core components (multi-probing, impact scoring, GTNet architecture, training objectives) in compact tables for quick reference.

- Discussed how presence of rare tokens can be handled and why pairwise ranking loss works well despite high sample complexity

- Discussed an error estimator for evaluating Chamfer distance

CoRGII addresses the growing need for scalable, structure-aware retrieval in massive graph corpora, offering an efficient and accurate solution that bridges the gap between dense neural graph representations and practical inverted indexing systems for neural graph search systems.

We are committed to incorporating these improvements into the final version to maximize the impact of our contribution.

---

### Decision · Program_Chairs · 2025-09-17

**Decision:**

Accept (poster)

**Comment:**

**Summary:** This paper presents CoRGII, a pipeline that combines contextual graph tokenization, differentiable discretization, and inverted indices for efficient graph retrieval. The method achieves strong accuracy - efficiency trade-offs and is supported by extensive experiments, ablations, and complexity analysis.

**Strengths:** The reviewers highlighted: (1) the importance of the problem; (2) a novel adaptation of classical IR techniques (contextual tokenization, differentiable discretization, and inverted indices) for graph retrieval; (3) producing strong empirical results over strong baselines with extensive ablations.

**Weaknesses:** Initial concerns included (1) unclear presentation; (2) missing complexity/runtime analysis, (3) reliance on IsoNet embeddings, (4) effects of discretization, and (5) robustness to rare tokens and noisy graphs.

The rebuttal and discussion resolved most of the reviewers' concerns. The authors clarified terminology, detailed the tokenizer architecture, and showed that CoRGII does not rely on pre-trained GNN embeddings. They added experiments on runtime efficiency, multiprobing, Chamfer error estimation, token interpretability, and rare-token handling, alongside clearer ablations. Reviewers acknowledged these clarifications, raised scores, and agreed that despite some remaining clarity issues (which the authors have agreed to resolve in the final version), the contribution is solid, novel, and impactful. The consensus is that this is a strong paper deserving acceptance at NeurIPS.